# Nano-biosupercapacitors enable autarkic sensor operation in blood

Yeji Lee [1,2,3], Vineeth Kumar Bandari [1,2,3 ✉], Zhe Li[1,2,3], Mariana Medina-Sánchez [3], Manfred F. Maitz [4], Daniil Karnaushenko[3], Mikhail V. Tsurkan [4], Dmitriy D. Karnaushenko [3] & Oliver G. Schmidt [1,2,3,5 ✉]

Today's smallest energy storage devices for in-vivo applications are larger than $3\,mm^3$ and lack the ability to continuously drive the complex functions of smart dust electronic and microrobotic systems. Here, we create a tubular biosupercapacitor occupying a mere volume of $1/1000\,mm^3$ (=1 nanoliter), yet delivering up to 1.6 V in blood. The tubular geometry of this nano-biosupercapacitor provides efficient self-protection against external forces from pulsating blood or muscle contraction. Redox enzymes and living cells, naturally present in blood boost the performance of the device by 40% and help to solve the self-discharging problem persistently encountered by miniaturized supercapacitors. At full capacity, the nano-biosupercapacitors drive a complex integrated sensor system to measure the pH-value in blood. This demonstration opens up opportunities for next generation intravascular implants and microrobotic systems operating in hard-to-reach small spaces deep inside the human body.

[1] Material Systems for Nanoelectronics, Chemnitz University of Technology, Chemnitz, Germany. [2] Research Center for Materials, Architectures and Integration of Nanomembranes (MAIN), Chemnitz University of Technology, Chemnitz, Germany. [3] Institute for Integrative Nanosciences, Leibniz IFW Dresden, Dresden, Germany. [4] Leibniz-Institut für Polymerforschung Dresden e.V., Dresden, Germany. [5] Nanophysics, Faculty of Physics, TU Dresden, Dresden, Germany. ✉email: vineeth-kumar.bandari@main.tu-chemnitz.de; oliver.schmidt@main.tu-chemnitz.de

Microelectronics has brought outstanding achievements to the field of intravascular implants, in vivo smart dust and small autonomous systems[1–7]. The miniaturization of microsystems progresses quickly but the demand for ever smaller energy storage devices that enable self-sufficiency and autarkic operation is increasingly more difficult to satisfy[8]. The most commonly available energy storage units at the submillimetre scale are micro-electrochemical capacitors (also termed micro-supercapacitors)[8,9], but they rely on non-biocompatible materials and self-discharge quickly making them unsuitable for in vivo applications. Biosupercapacitors (BSCs), instead, are fully biocompatible and can compensate the self-discharge behaviour via bioelectrocatalytic reactions[10]. The first BSC was demonstrated with a bacterial biofilm[11]. The energy was stored by combining redox enzymatic reactions with electrochemical reactions known as bioelectrocatalysis[10]. The bioelectrocatalytic reaction produces protons at the bioanode that are transferred and collected at the biocathode. In biological fluids, the generated protons can be used by living cells to perform metabolism[12] or by redox enzymes and glucose to trigger active catalytic reduction. In addition, these complex biological reactions release energy which nBSCs can harness to compensate for any self-discharge[10–13]. Much effort has gone into exploring and understanding the intricate pathways of this bioenhancement and self-discharge behaviour[14–18], but the side redox reactions leading to self-discharge are still true challenges. The low working potential (0.3–0.8 V) and large size (>3 cubic millimetres) of state-of-the-art BSCs[14–18] do the rest to deny these devices access to various small spaces in the human vascular system.

Here, we demonstrate a fully microsystem integrated, bioenhanced and robust nano-biosupercapacitor (nBSC) working in biological electrolytes (medical saline, blood plasma and blood). The small volume ($1 \times 10^{-3}$ mm$^3$ = 1nL) of the nBSC is achieved by self-assembling planar structures into a 3D compact tubular geometry, which in turn allows for stable operation under hemodynamic conditions with varying temperatures, pulsating blood flow and self-protection against external forces (up to 60 kPa). The energy storage and ion transport in the nBSC take place between two 100 nm thin flexible poly(3,4-ethylenedioxy-thiophene)-poly (styrenesulfonate) (PEDOT:PSS) redox- electrode layers through a 500 nm photopatterned poly(vinyl alcohol) (PVA) proton exchange separator and blood as the working electrolyte. The proton exchange separator and the compact "Swiss-roll" geometry are crucial elements to tackle self-discharge and miniaturization, respectively. The proton exchange separator acts as an active barrier for the protons generated at the bioanode, inhibiting the migration towards the biocathode, thus suppressing side redox reactions that lead to self-discharge[19,20]. The PVA proton exchange separator is key to retain high open voltages over 16 h and an average 40% enhancement in performance (capacity and energy storage). We passivate the nBSC with SU8 photoresist for quasi-electrical and ionic passivation enabling the device to operate at high working potential (1 V–1.6 V) without any gas evolution in biological electrolytic fluids (blood and blood plasma). We find that the capacitance of the nBSC varies as a function of the electrolyte pH. By integrating three charged nBSCs with a nBSC based ring oscillator, we realize a self-powered sensor for monitoring the pH of blood. This demonstration promotes nBSCs as excellent candidates for miniaturized biocompatible intravascular implants, in vivo smart dust[1–6] and microrobotic systems[7] with broad application potential in the personalized healthcare sector.

functional components such as current collectors, working electrodes, separator, passivation layer and electrolyte must be biocompatible. The physical dimensions should be compact and fit into the body's network of small blood vessels and fluidic channels withstanding the different physiological flows and pressures. The technology of choice for these requirements is the self-assembly of compact "Swiss-roll" Origami structures on a substrate surface. Using established on-chip fabrication processes[21,22], the Origami technique first layers down the required materials for the nBSC devices onto a polymeric thin film stack consisting of a sacrificial, hydrogel and polyimide layer (Fig. 1a and Supplementary Figs. 1–4). The device layers are then delaminated from their substrate by gradually dissolving the sacrificial layer. Driven by the expansion of the underlying hydrogel, the entire layer stack curls up (Fig. 1b) and self-assembles into an array of compact 3D devices with high accuracy and ~95% yield (Fig. 1c, d)[23]. The tuneable hollow core diameter (Supplementary Fig. 5) of the "Swiss-roll" nBSC provides a natural channel for the flow of aqueous biological electrolytes (Fig. 1e). And despite the quasi-electronic and ionic passivation of the device by SU8 photoresist, the PEDOT electrode still very well interacts with the electrolytes and their components due to the efficient fluid absorption of the PVA separator (Supplementary Fig. 6 and Note 1).

The "Swiss-roll" nBSCs were evaluated by a two-electrode measurement configuration in three different aqueous electrolytes (0.9% NaCl, blood plasma and blood). For NaCl and blood plasma, the cyclic voltammetry (CV) curves show quasi-rectangular shape (Fig. 1f), and the galvanostatic charge-discharge (GCD) graphs are approximately symmetric and linear in time (Fig. 1g), indicating typical electrochemical capacitive behaviour[24]. For blood, the CV curves deviate from the typical quasi-rectangular shape and the GCD curves show a slight asymmetric behaviour due to enhanced charge transfer at the electrode/electrolyte interface caused by bioelectrochemical reactions[10,13]. The nBSCs show a 40% increase in volumetric capacitance and energy density (see Fig. 1h and Supplementary Fig. 7). The enhancement of the electrochemical performance is attributed to the complex composition of blood. Blood consists of enzymes and living cells that produce bioelectrochemical reactions such as catalytic enzymatic reaction, cellular metabolism and glucose reduction creating a complex electrochemical environment. These reactions are absent in pure ionic electrolytes (e.g., 0.9% NaCl). The tubular nBSCs are capable of delivering an average volumetric energy density of ~90 nWh mm$^{-3}$ and a power density of ~32 µWmm$^{-3}$ at 50 nA in blood (Supplementary Fig. 7 and Note 2). The complex composition of blood and redox polymeric electrodes allow the nBSCs to work at a stable operating potential window of 1 V[25]. Moreover, the quasi-electronic and ionic passivation provided by the SU8 photoresist and PVA separator enable the nBSCs to reach a single device peak voltage of 1.6 V without any gas evolution in all aqueous electrolytes (Fig. 1i, Supplementary Fig. 8, 9, Note 3 and Supplementary Movie 1).

The nBSCs show excellent lifetime and structural integrity in blood retaining up to 70% of the initial capacitance with a coulombic efficiency of ~85% over 5000 cycles (Fig. 1j, Supplementary Fig. 10 and Note 4). GCD curves are measured for various currents (20, 40, 60, 80 and 100 nA), (Supplementary Fig. 11). At high current densities, fast kinetic capacitive processes take place at the electrode/electrolyte interface providing high driving power[26]. Compared to state-of-the-art BSCs[14–16], our nBSCs are more than three orders of magnitude smaller in size and operate in a wider voltage window (1 V– 1.6 V).

## Results

**Fabrication and electrochemical performance**. Nano-biosupercapacitors have to meet several demands. The materials and

**Bioenhancement**. Typical supercapacitors store energy at the electrode/electrolyte interface by migration of positively

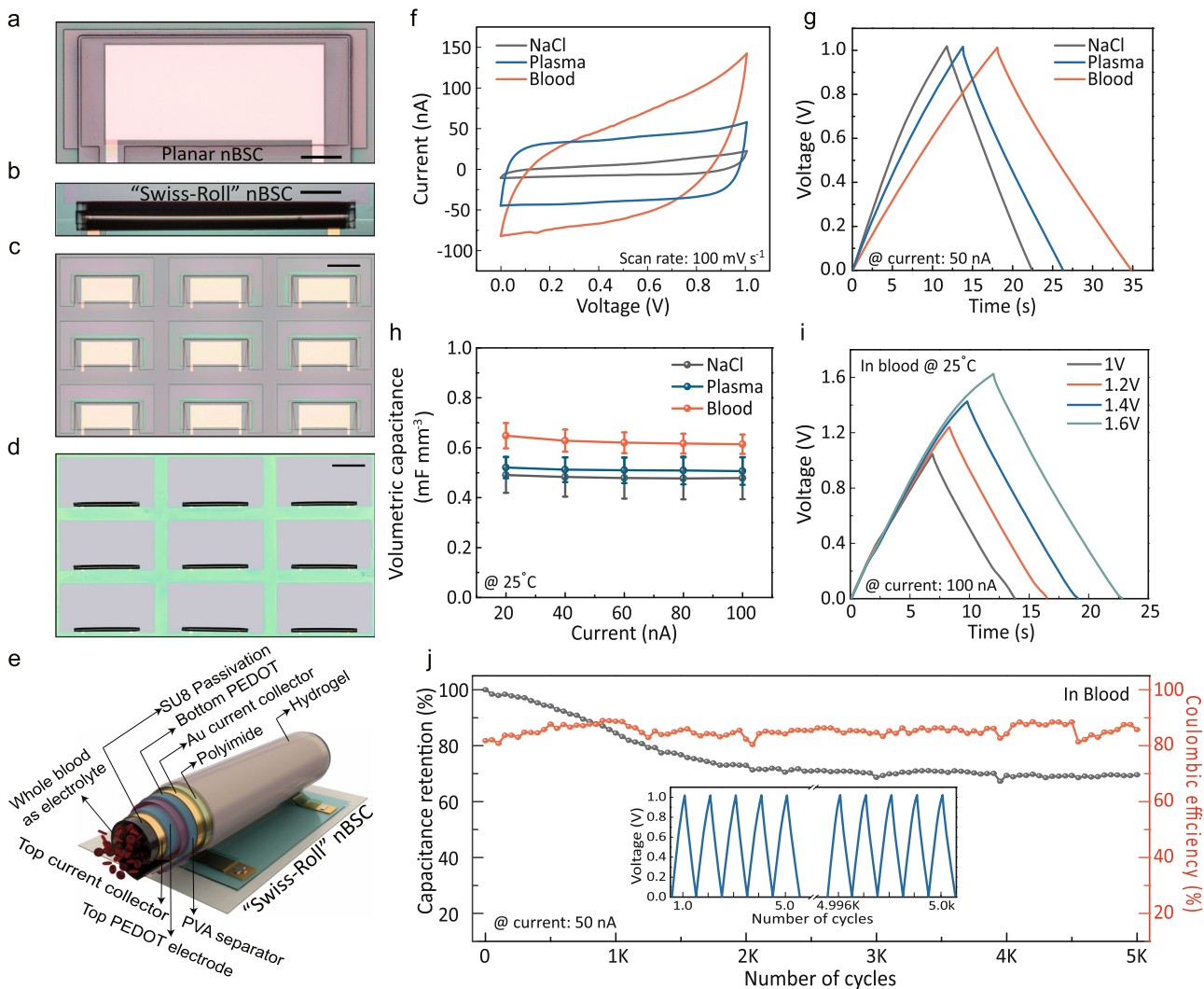

**Fig. 1 Fabrication and electrochemical performance of nBSC. a** Microscope image of a completed nBSC before rolling. **b** Microscope image of a "Swiss-roll" nBSC. Scale bar, 200 μm (**a**, **b**). **c** Microscope image of an array of nBSCs before rolling and (**d**) after self-assembly into tubular "Swiss-rolls". Scale bar, 500 μm (**c**, **d**). **e** Schematic illustration showing all active components of "Swiss-roll" nBSC with hollow core for the fluent flow of blood: polymeric rolling stack, gold (Au) top and bottom current collectors, PEDOT:PSS active top and bottom electrode, polyvinyl alcohol (PVA) proton separator and SU8 photoresist passivation. **f** CV curves of nBSCs in different biological electrolytes (NaCl, blood plasma, blood) at a scan rate of 100 mV s$^{-1}$. **g** GCD curves at an applied current of 50 nA. **h** Volumetric specific capacitance as a function of applied current in different electrolytes. **i** GCD curves at four different potential windows at 100 nA. **j** Capacitance retention and coulombic efficiency of a device in blood over 5000 cycles. (in Fig. 1h, error bars represent the variation in data over three measured devices. All measurements are performed under ambient conditions at 25 °C).

charged ions towards the negatively charged cathode and vice versa (Fig. 2a)[24,26], leading to a constant charge across the device under ideal conditions. However, leakage current, charge redistribution and faradaic side reactions result in self-discharge and short lifetime[27,28]. Charge redistribution and leakage current are often governed by defects and internal resistances but can be avoided by a carefully executed fabrication protocol. However, any faradaic side reaction, such as the production of water by oxygen reduction at the biocathode, causes inacceptable self-discharging of the device and remains an open challenge[29].

To suppress fast-self-discharge, a proton exchange separator (PES) can be employed[18,30,31]. The PES acts as an active barrier for the generated protons at the bioanode preventing the migration towards the biocathode, thus minimizing faradaic side reactions and leading to reduced self-discharge[19,20]. The PES of our nBSCs is synthesized by reacting 10% PVA with 0.003%

potassium-based photo-crosslink additives in distilled water[32]. The PVA based PES shows excellent electrolyte absorption properties while impeding the migration of active anions in the electrolyte and trap impurities[19,20] (see Fig. 2b), directly resulting in improved charge retention and reduced self-discharge over 1.5 h in 0.9% NaCl and blood plasma electrolyte. Biosupercapacitors can self-charge by utilizing redox-active components, such as metal centres in proteins, redox cofactors in enzymes, and low-molecular-weight natural redox mediators existing in organelles and active cellular metabolism in blood (Fig. 2c)[10–13]. To better understand the self-charging and bioenhancement behaviour, the nBSC electrodes are immersed and analysed in a three-electrode electrochemical measurement set-up filled with 0.9% NaCl and blood electrolyte with different working electrode configurations (see Fig. 2d). The evaluation of the electrochemical CV curves in Fig. 2e reveal prominent redox and strongly polarized peaks in blood stemming from the bioelectrocatalyic redox reactions[33].

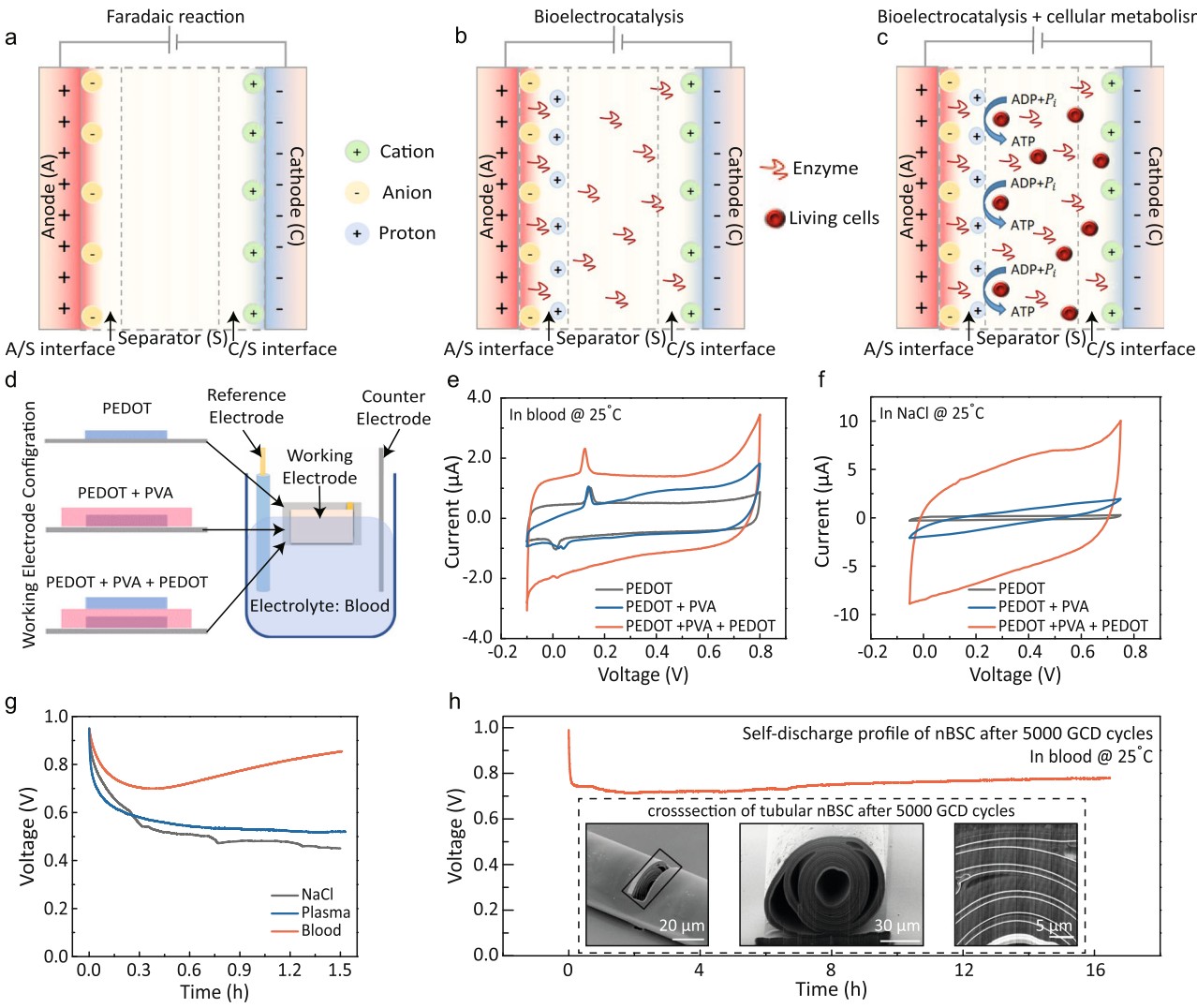

**Fig. 2 Mechanism of bioenhancement in a nBSC. a–c** Schematic illustration of ion transport and energy storage mechanism between two working electrodes in different electrolytes: **a** Faradaic reactions in presence of $Na^+$ and $Cl^-$ ions in medical saline. **b** Faradaic reactions coupled with bioelectrocatalysis due to the presence of enzymes in blood plasma electrolyte. **c** ATP synthesis coupled with bioelectrochemical reactions enhancing the electrochemical response of the nBSC in blood. **d** Schematic illustration of a three-electrode setup with a working electrode (the measured sample), reference electrode (Ag/AgCl) and counter electrode (platinum rod) to determine redox/catalytic reactions. **e** CV curve of various working electrode configurations measured in blood electrolyte at a scan rate of 30 mV s$^{-1}$. **f** CV curve of various working electrode configurations measured in NaCl electrolyte at a scan rate of 30 mV s$^{-1}$. **g** Self-discharge/self-charge of nBSCs under different electrolyte fluids for 1.5 h. **h** Self-discharge profile in blood after 5000x GCD cycles. Inset shows cross-sectional images of tubular nBSC after 5000x GCD cycles. (All measurements are performed under ambient condition at 25 °C.).

This is in strong contrast to the results obtained from the control measurement in 0.9% NaCl (see Fig. 2f) where such peaks are absent. As a result, the bioenhanced self-charging behaviour compensates the self-discharge and allows the nBSCs to retain ~1 V when operated in blood (see Fig. 2g). Output voltage and structural stability of the nBSCs were measured after 5.5 days long cycling (5000x) in blood. Even then, the nBSCs are capable of retaining 0.7 V over 16 h at room temperature (25 °C) with a mere electrolyte volume absorption of ~0.5 nl (which is 10 billion times smaller than the total blood volume in an average human body) (see Supplementary Fig. 12, Note 5 and Supplementary Movie 2). In addition, the nBSCs also show slow self-charging and no significant changes of the tubular structure (see Fig. 2h). The relatively low self-charge rate is caused by degrading enzymes and cells in the blood during the preceding measurement period of 5.5 days.

**Performance under physiologically relevant conditions**. To sustain the physiological activity of the human body, blood needs hemodynamic capillary action at certain temperatures and flow rates[34]. Any implantable device within the circulatory system must withstand these physiological conditions with a stable performance. To evaluate the electrochemical performance of nBSCs under physiological conditions, the devices were placed into a polydimethylsiloxane (PDMS)-based microfluidic channel (see Supplementary Fig. 13) with a diameter of 120–150 μm to mimic a blood vessel. The microfluidic channel is connected to a commercial syringe pump for flow rate control and the temperature is set by an attached Peltier element (see Supplementary Fig. 14).

Blood flow varies depending on vessel diameter and blood pressure. Typical flow rates range from 3 to 26 ml/min in arteries and from 1.2 to 4.8 ml/min in veins (for vessel diameters from 800 to 1.8 mm)[35]. We tested our devices in a temperature window

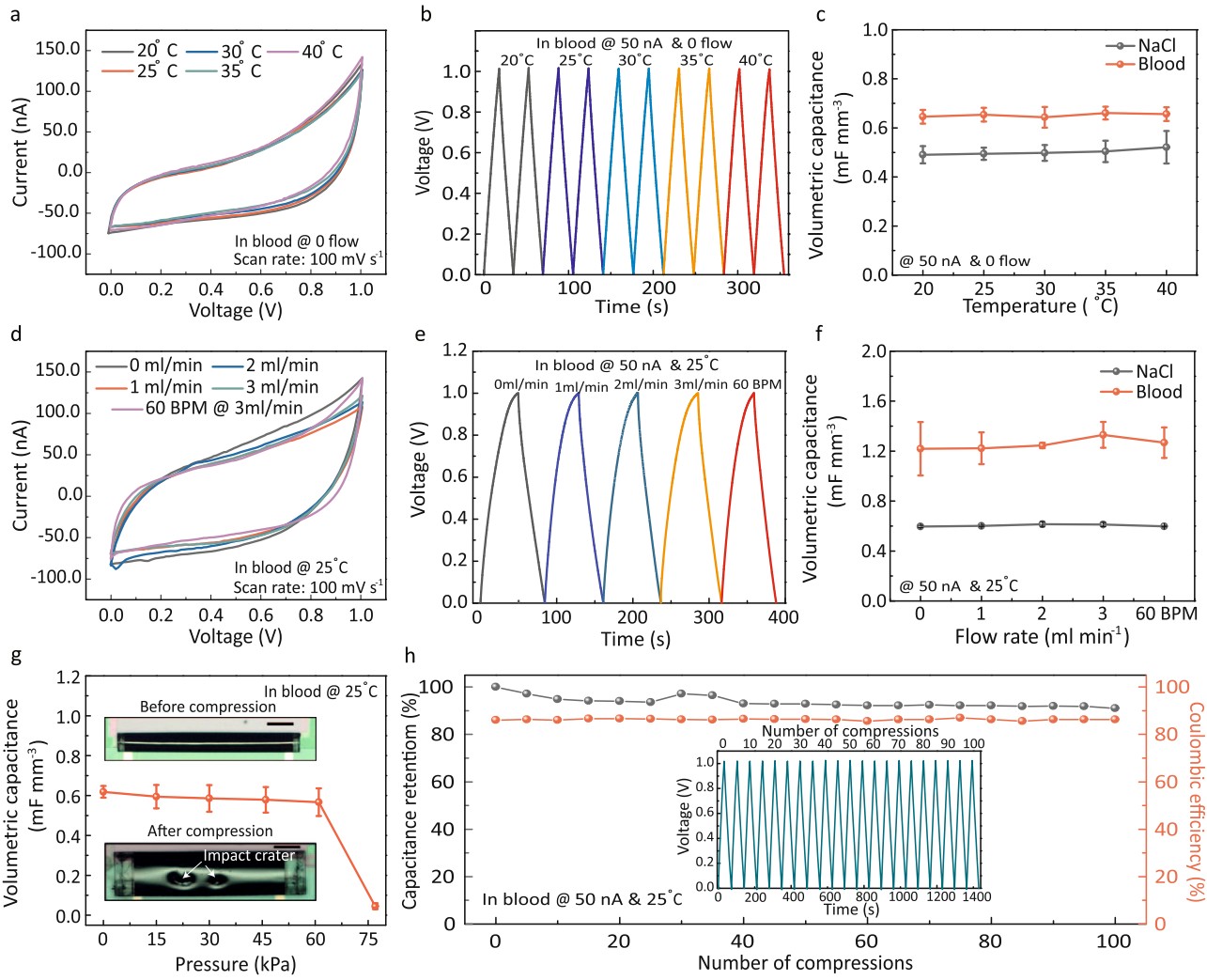

**Fig. 3 Performance of nBSC under physiologically relevant conditions. a–c** Temperature dependent measurements at static flow condition: **a** CV of nBSC at different temperatures with blood as electrolyte at a scan rate of 100 mV s$^{-1}$. **b** GCD curve at different temperatures with blood as electrolyte for an applied current of 50 nA. **c** Volumetric capacitance of nBSCs in different electrolytes as a function of temperature. **d–f** Flow dependent measurements at 25 °C: **d** CV of nBSC at different flow rates with blood as electrolyte at a scan rate of 100 mV s$^{-1}$. **e** GCD curve of nBSC at different flow rates with blood as electrolyte for an applied current of 50 nA. **f** Volumetric capacitance of nBSCs in different electrolytes as a function of various flow rates. **g** Volumetric capacitance of nBSCs in blood as a function of applied pressure. Inset shows nBSC before (top) and after (bottom) compression. Scale bars, 200 µm. **h** Capacitance retention and coulombic efficiency of a device under repeated compressions at 15 kPa over 100 cycles. Inset shows 100x GCD cycles (each GCD curve was measured after 5 compression cycles) of a nBSC subjected to 15 kPa. (in Fig. 3c, g, the error bars represent the variation in data over three measured devices.), (in Fig. 3f, the error bars represent the variation in data over two measured devices.).

from 20 to 40 °C and the flow rate of the electrolyte was changed from 0 ml/min to 3 ml/min at 60 beats per minute (BPM). CV curves (Fig. 3a), GCD curves (Fig. 3b) and electrochemical impedance spectra (EIS) (Supplementary Fig. 15) were recorded for various temperatures. Similar to Fig. 1f and g the CV and GCD curves exhibit polarised quasi-rectangular and slightly asymmetric triangular shape respectively (Supplementary Fig. 16). EIS over a frequency range from 1 Hz to 1 kHz at different temperatures shows a quasi-vertical character with respect to the −Z″ axis in the low-frequency region, which confirms the pronounced capacitive behaviour (Supplementary Fig. 15). Overall, the nBSCs behave well and are stable over the measured temperature range confirming the 40% performance enhancement for the devices operating in blood (Fig. 3c). Blood circulates and pulsates in the vascular system at different flow rates. The CV (Fig. 3d) and GCD curves (Fig. 3e) as a function of electrolyte flow rate exhibit excellent stability with only negligible changes. As shown in Fig. 3f, the nBSCs are able to retain their initial

capacitance in blood and show a ~ 100% enhanced performance over devices operating in NaCl. The more pronounced asymmetric behaviour (see Fig. 3e and Supplementary Fig. 17) and the additional ~60% enhancement of volumetric capacitance in Fig. 3c are a result of oxygen plasma treatment required to bond the microfluidic chip to the nBSC sample. The plasma treatment generates organic radicals that lead to the enhancement of the electrochemical performance of the nBSC (see Supplementary Fig. 18 and Note 6). As shown in the two-phase flow measurement setup (Supplementary Fig. 19 and Supplementary Movies 3–6) nBSCs alter the fluid flow rate and local fluid pressure. The target vessel must be selected such that the nBSC does not block the vessels by more than 30%. Any blockage over 30% would cause a dramatic decrease in blood flow velocity and increase in local blood pressure (see Supplementary Fig. 20–22, Supplementary Movie 7-8 and Note 7) leading to vascular remodelling and stenosis[36]. Although biofouling was not observed around the nBSCs during the short term (5.5 days)

in-flow GCD cycling, anti-biofouling strategies over the whole operation lifetime need to be implemented in future application scenarios to prevent vascular blockage.

The hemodynamic conditions of blood vessels are governed by hydrostatic pressures (2.0–5.3 kPa)[37], osmotic pressures (3.3–4kPa)[38], and external pressures exerted by vigorous intramuscular contractions (up to 30kPa)[39,40]. The tubular nBSCs were subjected to pressures from 15.4 kPa to 76.8 kPa via external compression (Fig. 3g, Supplementary Fig. 23) and the capacitance was measured as a function of applied pressure in real-time (Fig. 3g). The tubular geometry of the nBSCs provides excellent radial flexibility and mechanical stability against these forces. As shown in Fig. 3h and Supplementary Fig. 24a, the devices maintain very good charge/discharge characteristics under repeated compressions up to 15 kPa with only a small decrease in capacitance retention. Moreover, under reduced pressure of 5 kPa the device can withstand over 200 cycles with no significant change in neither capacitance retention nor coulombic efficiency (see Supplementary Fig. 24b).

The influence of nBSCs on cell proliferation, inflammation and thrombosis was investigated. Cell proliferation and viability studies were performed by incubating nBSCs with madin-darby canine kidney (MDCK) cells for a duration of two days. The absorbance spectrophotometry at 490 nm of the incubated samples demonstrates that all active materials as well as the complete tubular nBSCs do not show any significant effects on cell proliferation and viability when compared to the cell culture control in a tissue culture polystyrene (TCPS) (Supplementary Fig. 25). In addition, the nBSCs were incubated in human whole blood treated with 1.5 U/ml heparin for two hours to examine possible effects on inflammation and thrombosis of blood[41]. No hemolysis was induced by the nBSC, and as shown in Supplementary Fig. 26, inflammation reactions and blood clotting are low to moderate when compared to the control samples, implying that the nBSCs are safe to use and do not cause any significant cyto- and hemotoxic behaviour. Cyto- and hemotoxic behaviour could even be improved by systemic anticoagulation or surface-immobilized anticoagulants.

**Self-powered monolithic pH sensor system.** The potential of hydrogen (pH) of blood is controlled and maintained by constant production and consumption of acids and bases through chemical reactions during cellular activity[42]. Under normal conditions, the pH is in the range of 7.2–7.5. However, uncontrolled cellular mitosis around malignant tumours causes aerobic metabolism and excess carbon dioxide ($CO_2$) leading to a change in blood pH to around 6.4–7.0[43,44]. The detection of blood pH in real-time would therefore be a great help to predict tumour formation and proliferation.

The capacitance of the nBSCs in artificial plasma varies as a function of electrolyte pH with a sensitivity of $5 \pm 0.5\,\mu F\,mm^{-3}$ per acidic pH and $2 \pm 0.4\,\mu F\,mm^{-3}$ per basic pH (Supplementary Fig. 27). Changes in capacitance can be converted to a frequency modulation by a ring oscillator (with frequency inversely proportional to capacitance)[45]. A ring oscillator uses an odd number of inverters to generate a non-sinusoidal signal. The frequency of the non-sinusoidal signal is controlled by the delay time caused by parasitic capacitances between input and output stages[46]. Here, the last output stage of a five-stage ring oscillator circuit (Fig. 4a) is integrated with the pH-sensitive nBSC leading to an output frequency modulation as a function of electrolyte pH. The pH sensor based on the nBSC and ring oscillator is fabricated on the polymeric platform (see Fig. 4a) to assemble the device into a 3D tubular geometry (see Fig. 4b). The hollow inner core of the tubular sensor serves as a channel (Supplementary

Fig. 28) for flowing artificial plasma enabling particularly efficient pH detection. The tubular pH sensors are then powered by three on-board nBSCs connected in series (see Fig. 4a, b) and charged to ~3 V (Supplementary Figs. 29, 30 and Note 8). The three nBSCs connected in series can charge up to 3 V while maintaining contact with the same electrolyte due to the quasi-electronic and ionic isolation by the SU8 photoresist passivation and the PVA separator (Supplementary Fig. 31 and Note 9). The system is subjected to different electrolyte (artificial plasma) pH ranging from 1 to 7 (DI water used as calibration) for autarkic pH sensing (Supplementary Fig. 32 and Note 8). The change in volumetric capacitance as a function of pH is measured by the output frequency modulation of the ring oscillator (Fig. 4c). Under dry and static conditions, the pH sensor has an output frequency of 9.4 kHz (Fig. 4d). The autarkic tubular pH sensor shows an increase in frequency for basic pH and a decrease in frequency for acidic pH (see Fig. 4d) with a relative frequency change of $-2.5$ E-2 $\pm$ 1.9 E-4. Figure 4d and e shows the frequency modulation and output voltage swing of the devices between 9.2 kHz to 7.7 kHz (pH ranging from 1 to 7 (DI water)). Finally, the sensor system was able to operate in blood plasma and blood with a relative frequency change of $-2.7$ E-2 $\pm$ 2.0 E-2 per pH (Supplementary Fig. 33 and Note 10) and an output frequency of 6.7 kHz (Supplementary Fig. 32), respectively. The complex composition of biological fluids causes broadening of the spectral response with several overlapping frequency peaks (Supplementary Fig. 33 inset and Note 10) leading to larger error bars in the relative frequency change.

## Discussion

In conclusion, we have created self-charging nano-biosupercapacitors a thousand times smaller than a cubic millimetre. The devices are compatible to the hemodynamic conditions of the vascular system and show stable performance in energy storage and power output. The devices can be combined with complex microelectronic circuitry and measure the local pH value in blood potentially uncovering the treacherous formation of cancer cells. In combination with surface electrode functionalization, selective sensing of glucose, ascorbic acid, uric acid, dopamine, and lactate could indicate other potential diseases[47–49]. In order to translate the nBSC devices into real biomedical applications several challenges need to be addressed: The ability to charge the nBSCs in vivo without tether; device encapsulation against immunoreactions and biofouling caused by proteins, cells, bacteria and others; a smart way to implant the nBSCs in a vascular network with minimum invasion; possible biodegradation of non-functional devices. Overcoming these challenges by the materials versatility, the design flexibility as well as the extreme integration capabilities of on-chip manufacturing render these devices particularly promising microsystems for self-powered biomedical implants[1–6], smart cardiovascular stents[50] and autarkic motile microelectronic systems[7]. Once there is energy on board, it will be interesting to fathom out how such devices can transfer physiological information from deep inside the human body to the outside world. A particular interesting piece of information would be the mere signal of the position of the device, as deep-tissue tracking of tiny motile objects represents one of the most challenging tasks in the field of biomedical microrobots[51].

## Methods

**Preparation of polymeric layer stack.** For the preparation of the tubular nBSCs, a stack of polymeric layers (SPL) was first deposited consisting of sacrificial, hydrogel and polyimide layers. The syntheses of each functional component of the PRS has been reported in previous works[20,21]. The fabrication of the SPL starts by spin-coating a ~600 nm thick sacrificial layer (La-AA) on a Si/SiO$_2$ substrate (O$_2$

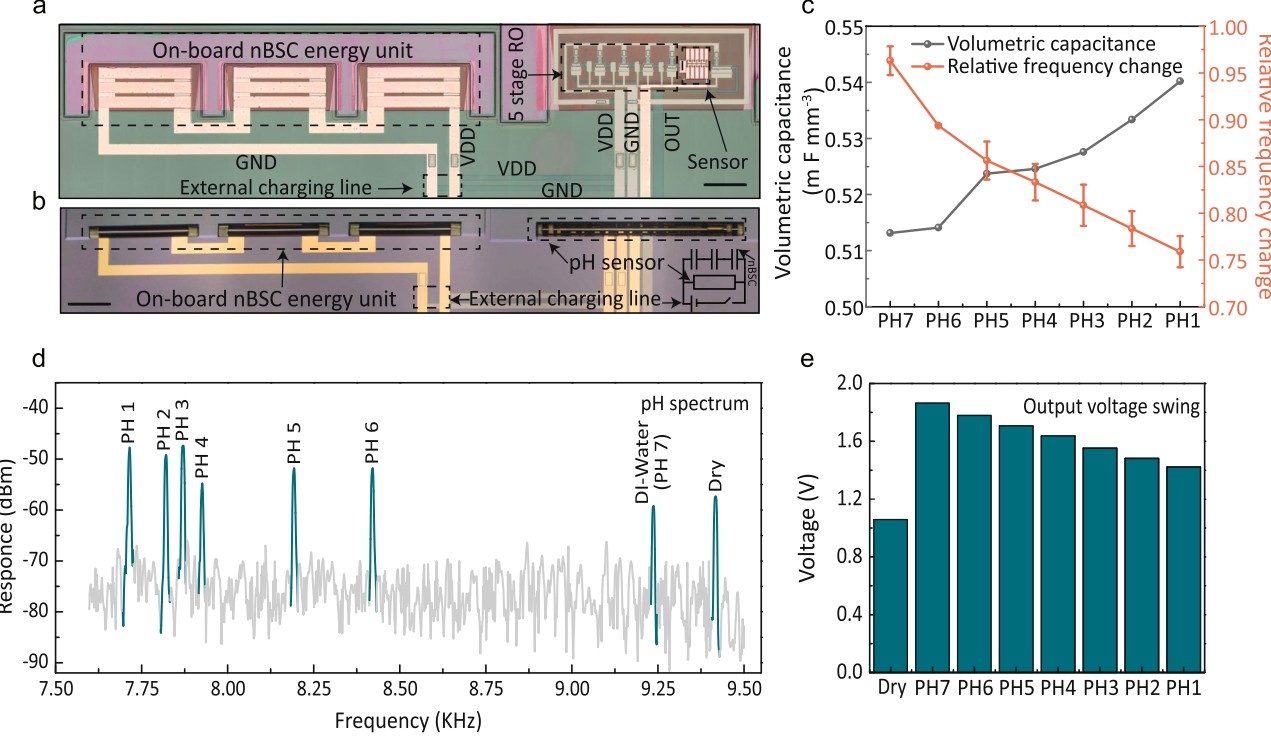

**Fig. 4 nBSC as a self-powered pH sensor. a** Microscope image of pH sensor with all integrated components (ring oscillator (RO) with nBSC sensing element connected to nBSC energy storage via drain-to-drain voltage (VDD), ground (GND)) before roll-up and (**b**), after roll-up, inset shows the operational circuit. Scale bars, 200 μm (**a**, **b**). **c** Volumetric capacitance and relative frequency change of nBSC as a function of electrolyte pH. **d** Frequency spectral response of the nBSC based pH sensor at various electrolyte pH. **e** Output voltage swing of a five-stage oscillator working at 3 V DC input provided by the on board nBSC as a function of electrolyte pH. (in Fig. 4c, error bars represent variation in data over three measured devices. All measurements are performed under ambient conditions at 25 °C in artificial plasma flow of 2 ml/min).

plasma-cleaned). The sacrificial layer was spin coated at 4,000 RPM for 90 s on the substrate followed by a soft bake at 40 °C for 10 min. After soft bake, the sacrificial layer was patterned by UV-exposure (365 nm, 15 Wcm$^{-2}$) through a photomask using a SÜSS MJB4 mask aligner (Karl Süss). The exposed pattern was developed in DI- water for 90 s and then hard baked at 220 °C for 30 min.

After the completion of the sacrificial layer, hydrogel solution was spin-coated at 5000 RPM for 90 s and soft baked at 40 °C for 5 min. The hydrogel layer was UV-exposed through a photomask (365 nm, 15 Wcm$^{-2}$) for 65 s and the pattern was developed in diethylene glycol mono ethyl ether (DEGMEE, from Alfa Aesar) for 30 s followed by a hard bake at 220 °C for 30 min to achieve a ~1 μm thick film on top of the sacrificial layer.

Finally, the fabrication of the SPL is completed by coating the surface with a ~700 nm thick polyimide layer at 6000 RPM for 90 s followed by a soft bake at 50 °C for 10 min. The samples where patterned by UV-exposure (365 nm, 15 Wcm$^{-2}$) and developed in a solution prepared by mixing 1 part (vol/vol) of ethanol, 2 parts (vol/vol) of DEGME and 4 parts (vol/vol) of N-Ethyl-2-pyrrolidone for 10 s. After patterning, the layers were stabilized by a hard bake at 220 °C for 30 min.

**Fabrication of nBSCs**. After the fabrication of the SPL, the functional layers (bottom/top current collector, symmetric working electrode, proton exchange membrane and passivation) of the nBSCs were deposited in a symmetric sandwich architecture on the SPL (Supplementary Figs. 1, 2 and 3).

*Current collectors*. Standard photolithography was used to negatively pattern AZ 5214E photoresist (from Micro Chemicals) on the polymeric layer stack. After patterning the photoresist, 10 nm Cr and 50 nm Au were deposited at 0.5 Å s$^{-1}$ in an e-beam evaporator (Creavac). Finally, a lift-off process was performed in acetone and isopropyl alcohol to remove the photoresist and residue metal layer.

*Working electrodes*. The active working electrode material was prepared by filtering PEDOT:PSS (CLEVIOS™ F HC Solar) through a 0.45 μm PVDF filter. The filtered PEDOT:PSS solution was spin-coated at 3,000 RPM for 60 s and baked at 125 °C for 20 min to deposit a ~100 nm thick film. The samples were then placed into an e-beam evaporator to deposit a 200 nm thick Ag layer at 0.7 Å s$^{-1}$ which serves as an etch mask. Following the Ag deposition, AZ 5214E photoresist was spin coated and subjected to positive photo patterning. The Ag layer was then etched in solution of 1 part (vol/vol) of DI water, 3 parts (vol/vol) of phosphoric acid (85%)

and nitric acid (65%), and 23 parts (vol/vol) of acetic acid (100%). Following the Ag etching, the PEDOT:PSS was etched using O$_2$ plasma (TEPLA) at 400 W for 2 min. The samples were cleaned in acetone and isopropyl alcohol to remove the residue of the photoresist and the Ag layer was removed completely using the above etching solution (The topography can be seen in Supplementary Fig. 4).

*Proton exchange separator membrane*. The separator membrane was synthesised by mixing 1 g poly (vinyl alcohol) (PVA, from Sigma Aldrich) and 5 mg potassium dichromate in 10 ml DI-Water at 80 °C for 12 h. The solution was cooled and spin coated at 9,000 RPM for 60 s by using a 0.45 μm PVDF filter and soft baked at 40 °C for 4 min to deposit a ~500 nm thick film. The PVA films were then photo-exposed (365 nm, 15 Wcm$^{-2}$) for 90 s through a photomask and the separator was patterned by developing the samples in DI water for 20 s (The topography can be seen in Supplementary Fig. 4).

*Passivation*. To protect the devices from the corrosive liquids during the rolling process, a ~600 nm thick SU8-2000.5 photoresist (from Micro resist technology) was spin-coated at 4500 RPM for 60 s and soft baked at 95 °C for 4 min. The samples where then patterned to remove the SU-8 to form a trench for rolling by UV-exposure (365 nm, 15 Wcm$^{-2}$) for 65 s through a photomask and a post exposure bake at 95 °C for 2 min was done to accelerate the crosslinking process of the exposed area. Finally, the trench was formed by developing the samples in mr-Dev 600 (from Micro resist technology) for 60 s. The SU-8 was introduced to suppress water splitting.

*Self-assembly of tubular nBSCs*. The rolling process was controlled by selectively etching the sacrificial layer in a solution of 0.1 M sodium diethylene triamine penta acetic acid (DTPA, from Alfa Aesar) (pH level adjusted to 9 by sodium hydroxide). After completing the rolling process, the rolled-up nBSCs were immersed in DI water for 30 min, with slight agitation, to wash away the residual rolling solution.

**Structural characterization and analysis of nBSCs**. The morphologies of tubular nBSCs were inspected by SEM/FIB (TESCAN) at 30 keV Ga$^+$ and optical micro-scopy (Olympus BX5). The thickness of each layer was measured from the cross-sectional SEM images, and a surface profiler (Veeco Deketak 8). The surface roughness and topography were recorded by AFM (Bruker).

**Electrochemical performance of nBSCs**. All electrochemical measurements were performed using a µAutolab Type III potentiostat (from Metrohm with internal impedance of >100 GOhm).

*Three-electrode configuration.* The system consists of a working electrode (the measured sample), reference electrode (Ag/AgCl) and counter electrode (platinum rod), which are all connected to a potentiostat. The measurements were performed in 0.9% NaCl and blood at room temperature (25 °C). To physically connect to the Potentiostat (µ-Autolab), a micro-probe station (cascade Microtech) was used. A microscope was used to align the micro-probe of the probes station to the bond pad area of the device. Finally, the micro-probes were connected to the Auto-lab work station using BNC connectors.

**Cell viability test**. To check the toxicity of the employed materials, MTS (3-(4,5-dimethylthiazol-2-yl)-5-(3-carboxymethoxyphenyl)-2-(4-sulfophenyl)-2H-tetrazolium) a cell proliferation assay (Abcam, ab197010) was performed. The method is based on the reduction of MTS tetrazolium by viable cells generating a coloured formazan product. The produced formazan can be quantified by measuring the absorbance at 490 nm.

In total, 8 samples at different preparation steps were analyzed. All samples had a dimension of $10 \times 10$ mm. As control sample tissue culture polystyrene (TCPS) was used. All samples were placed in 12 well plates with cell repellent surface, to avoid cells adhering on the surface of the plate and growing only on the sample. In total, 1.5 ml Medium (Dulbecco's Modified Eagle's Medium - low glucose + 10% FCS + 1% Penicillin/Streptomycin) and 30,000 cells (MDCK-C7) were added and cultured for 48 h at 37 °C and 5% $CO_2$. For every sample, three replicates were used. After 48 h of incubation, the samples were transferred into new well plates with new medium. 150 µl MTS-solution was added into each well. The plates were incubated for 2 h at 37 °C and 5% $CO_2$. After incubating the samples for 2 h, the plates were shaken briefly on a shaker and absorbance was measured by using a plate reader at 490 nm.

**Blood clotting and inflammation assays**
*Heparin coating for blood anticlotting.* To improve the heparin immobilization, tubular nBSCs were first subjected to aluminium oxide ($Al_2O_3$ ~5 to 10 nm) deposition. Oxygen plasma was applied (2 min) to activate the surface molecules of the deposited oxide. Meanwhile, APTES (2%) solution was prepared with 5% DI water and 93% ethanol. The samples were transferred to the APTES solution after plasma activation and incubated for 1 h at room temperature to create amine bonds on the object surface. The samples were rinsed afterwards with ethanol and then with PBS. Then, N-ethyl-N-(3-dimethylaminopropyl)-carbodiimide hydrochloride (EDC) at a concentration of 5.712 mM, and active ester compound N-hydroxy sulfosuccinimide (NHS) at a concentration of 2.856 mM, were used for coupling the carboxyl groups from the heparin (0.714 mM) to the amino groups from the nBSC surface, forming covalent bonds (during 2 h at room temperature). Finally, the samples were washed with PBS containing 50 mM glycine for quenching residual reactivities and DI water for further experiments.

*Blood anticlotting experiments.* The abovementioned protocol to coat the prepared structures with heparin was applied to the following samples: substrate, planar and rolled-up nBSCs. The whole human blood incubation assay was performed by using quasistatic incubation chambers, as described in previous works[52]. Samples were incubated with whole human blood treated with 1.5 U/ml heparin for 2 h without air interface and under constant overhead rotation. Different biomarkers concerning coagulation and inflammation were determined. Blood cell counts were performed on an automated cell counter (AcT diff, Coulter, Germany) to determine the decrease in cell number. The prothrombin F1 + 2 fragment and complement fragment C5a were measured with a commercial ELISA kit (Enzygnost F1 + 2 micro, Siemens Healthcare, and C5a ELISA, DRG Instruments). To determine the formation of platelet-granulocyte conjugates as a marker of blood platelet activation, the blood sample was stained with CD15-PE and CD41a-FITC (both Becton Dickinson) and analyzed by flow cytometry (LSR Fortessa, Becton Dickinson) following a lyse-no-wash protocol. CD41a presenting granulocytes are regarded as conjugates with platelets. To determine the CD11b expression level, the blood sample was stained with CD11b-PacificBlue (Biolegend) and analyzed following a lyse-no-wash protocol. The result was obtained by flow cytometry (LSR Fortessa). The samples were also fixed using formalin at 2% in PBS, and stained with DAPI and DiOC6a for fluorescence microscopy and SEM analysis (Supplementary Figs. 25 and 26).

**Fabrication and integration of microfluidic channel**. Silicon elastomer and curing agent (from VWR™) were mixed in a weight ratio of 10:1, the solution was placed in a desiccator to remove air bubbles that were generated during the mixing process. In parallel, a 150 µm thick SU8-3005 photoresist (from Micro resist technology) was spin coated on a cleaned Si/SiO2 wafer and exposed to standard lithography to achieve the master mold of the microfluidic channel. The bubble free PDMS was casted over the SU8 photoresist mold and was cured at 65°C for 6 h. After the curing process, the PDMS was peeled-off from the mold and was bonded to the tubular nBSCs using $O_2$ plasma bonding (Supplementary Fig. 13).

**nBSCs based pH sensor**
*Three-stage ring oscillator.* To create the gate of a thin film transistor (TFT), standard photolithography was used to achieve a negative pattern on the polymeric layer stack. After patterning the photoresist, 25 nm Cr was deposited at 0.5 Å s⁻¹ by e-beam evaporation (Creavac). Finally, a lift-off process was performed in acetone and isopropyl alcohol to remove the photoresist and residue metal layer.

In total, 6 nm hafnium dioxide ($HfO_2$)/3 nm aluminium oxide ($Al_2O_3$)/6 nm $HfO_2$ were successively deposited by atomic layer deposition (Cambridge nanotech, Savannah s100) at a substrate temperature of 220 °C to form the gate-dielectric layer. $HfO_2$ and $Al_2O_3$ films were grown by precursors of tetrakis (ethylmethylamino)hafnium (TEMAH), trimethylaluminum (TMA) and $H_2O$ respectively (from Strem Chemicals, Inc.).

Following the gate dielectric layer, 20 nm ZnO as active semiconductor layer was deposited using a dimethyl zinc (DMZ) precursor and $H_2O$ by atomic layer deposition (ALD) at a substrate temperature of 110 °C. The sample was then annealed in vacuum at 300 °C for 2 h with a temperature ramp rate of 1 °C/min.

The source and drain layers were formed by depositing 80 nm Ti and 30 nm Au at 0.25 Å s⁻¹ by e-beam evaporation (Creavac) on a negatively patterned layer. After lift-off in acetone and isopropyl alcohol, the active transistor with a width to length ration of W/L = 100/5 µm and the load transistor with W/L = 10/5 µm were completed. After source and drain formation, samples were annealing at 300 °C for 2 h to decrease contact resistance.

Finally, the device was passivated by spin coating a ~1 µm thick polyimide layer at 2500 RPM for 60 s which was soft baked at 50 °C for 10 min. After soft bake, a 200 nm $SiO_2$ layer was deposited at 0.5 Å s⁻¹ by e-beam evaporation (Edwards auto 500 FL). The respective transfer characteristic are shown in Supplementary Fig. 34, and the devices after several consecutive fabrication steps are shown in Supplementary Figs. 35 and 36.

For the sensing element and energy storage system, the nBSCs were fabricated using the above fabrication protocol in the respective locations on the chip as shown in the Supplementary Figs. 37 and 38.

**Use of electrolytic fluids**
*Human blood.* Used for blood clotting and inflammation assays.

*Bovine blood.* Used in all CV, GCD, self-discharge, three-electrode and PH sensor measurements (including all temperature and flow tests). Was also used in sensor experiments.

*Blood plasma.* Extracted from bovine blood and used in CV, GCD and self-discharge measurements.

*Cell culture media.* Used for incubating MDCK-C7 cells for viability and proliferation test.

*Medical saline (0.9% NaCl from B. Braun Melsungen AG).* Used in all CV, GCD, self-discharge and three-electrode measurements (including all temperature and flow test).

*Artificial plasma electrolyte for pH sensor.* Artificial plasma reproducing the chemical composition of human blood plasma[53] (Supplementary Table 1) was used in the sensor experiments. pH levels of the artificial plasma electrolyte were adjusted by adding 1 M hydrochloric acid (for acidic pH) and sodium hydroxide (for basic pH).

*Blood plasma electrolyte for pH sensor.* The pH of blood plasma was altered by adding and adjusting the concentration of citric acid to achieve a blood plasma pH from 1 to 7.

## Data availability
The data that support the findings of this study are available on request from the corresponding authors.

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

## Acknowledgements

We thank F. Hebenstreit for the help with the sample preparation and toxicity test, K. Leger for preparing the polymeric rolling stack, C. Becker for the calibrations and characterization of the polymeric materials and Y. Hong for the support in fabrication. We thank C.N. Saggau for performing the absorption spectroscopy and S. Andleeb for technical help with scanning electron microscopy. We acknowledge P. Plocica, S. Nestler, and R. Engelhard for technical support and we thank M. Zhu for helpful discussions. M.M.-S. acknowledge the European Research Council (ERC) under the European Union's Horizon 2020 research and innovation program (grant agreement No. 853609). D.K. acknowledges financial support by the German Research Foundation (KA5051/1-1). O.G.S. acknowledges financial support by the Leibniz Program of the German Research Foundation (SCHM 1298/26-1).

## Author contributions

V.K.B. and O.G.S. conceived the idea and supervised the work. With help from Z.L., Y.L. and V.K.B. performed all steps from sample preparation to device measurements. Y.L., V.K.B. and O.G.S. analysed the data and wrote the manuscript with input from all authors. M.M.-S. M.M., M.V.T. performed and analysed the cyto- and hemocompatibility tests. D.D.K. and D.K. prepared the materials for the polymeric layer stacks.

## Funding

## Competing interests

The authors declare no competing interests.

## Ethics statement

The work with human blood has been approved by the ethics committee of the Sächsische Landesärztekammer under the license "Az. EK-BR-24/18-1". Blood samples were collected with the informed consent of donors

**Additional information**

