## [Peer Review File · Nature Communications]

Reviewers' Comments:

Reviewer #1:

Remarks to the Author:

This manuscript describes the preparation and characterisation of supercapacitor devices that are composed of alternating layers of PEDOT-PSS that are separated by a PVA proton exchange layer. The method of preparing the devices is novel, relying on the self-assembly of the device into what is aptly described as a "Swiss roll" arrangement. This allows a device to be made that possesses the shape of a tube, one that is sufficiently small to be able to fit into blood vessels. Detailed characterisation of the devices in terms of their properties and performance is presented and described in the figures, text and SI. Of note is the ability of the device to retain good capacitance and coulombic efficiency for 5000 cycles (fig 1). The device also possesses good biocompatibility properties and as described (lines 225 et seq.) no significant issues with inflammation etc. The results obtained from the characterisation of the devices are promising and are indicative of a device that could be deployed and used in measurements for clinical applications. Overall the level of detail on the preparation and characterisation of devices is of a very good quality. The devices are incorporated into a self powered pH sensor, one that can measure changes in pH over the range 1-7 (fig 4e). This represents a novel and a significant advance in making measurements of this type.

One aspect that is unclear is how the current collectors are physically connected to the potentiostat? Details of this are not evident in the methods and need to be. Its also unclear as to how the device is charged for use? The device clearly operates as a supercapacitor but e.g. in figure 4 c, are the capacitors charged prior to use?

The properties of the devices in blood are enhanced in comparison with 0.9% saline, however the explanation for this enhancement is not clear. The text "Blood consists of enzymes and living cells that produce strong bioelectrocatalytic reactions)" line 107 is correct in itself but does not apply clearly here. The electrode layers in the devices are enclosed and only exposed at the edge (the side view of the swiss roll) as in fig 2h. Its not clear that bioelectrocatalytic reactions can occur as such reactions require an electrode surface that usually has to undergo modification to enable electrocatalytic reactions to occur (and by definition electrocatalytic implies electron transfer). Reactions such as those that utilise ADP/ATP do not involve electron transfer, typically this requires the use of NAD(P) cofactors and/or redox active enzymes. This aspect of the manuscript needs to be addressed and a more detailed explanation of the effect in blood needs to be provided before the manuscript can be accepted for publication. An additional comment is that the manuscript would benefit from a discussion on the size of the devices in terms of the radius of the channel, ca 30 μm . Was this radius selected or a consequence of the method of preparation? Can the size be altered and if so over what range? Does this size exclude cells (RBCs have a radius of ca 6 μm and should be able to enter the channel).

In summary the manuscript describes a novel device that has significant potential for applications in clinical measurements, specifically in blood. Its of interest to a broad readership such as that of this journal. Prior to recommending it for publication the questions above on the modus operandi of the device in blood need to be addressed.

Reviewer #2:

Remarks to the Author:

In this interesting work, Lee et al. report on the development of a roll-up micron-sized biosupercapacitor with enzyme-aided reaction to combat the self-discharge issue commonly faced by miniaturized supercapacitors. To demonstrate the application of such energy storage device, the authors developed an integrated pH sensing system powered by such nano-biosupercapacitor. The work is novel and the study is well done and of high quality. However, the visionary concept proposed

by the authors has some limitations towards practical biomedical application, which should be addressed and discussed. In particular:

1. As we know, the stability window in aqueous environment is only 1.21 V. The operation of the supercapacitor at 3 V is out of the stability window of the blood and may cause gas evolution. How is the device addressing this issue? Furthermore, as the BSC requires direct contact with the blood to maintain the voltage, how can three BSCs connect in series while sharing the same electrolyte?
2. The roll-up device occupies a significant amount of volume in the blood vessel, and can cause a significant increase in local blood pressure and may cause a reduction in flow rate. The authors propose to insert the device inside the blood vessels. Some comments on this concern would be helpful.
3. How will the rolled-up BSC be charged while implanted within the blood vessel? Is it through a wired connection? If so, how to connect to the BSC through the blood vessel?
4. The blood vessel dilates and contracts for tens to hundreds of times per minute, which might cause repeated mechanical deformations on the BSC device. Figure 3h have shown the stability of the BSC in 75 cycles of compression, but its stability through more deformation cycles are needed to ensure its mechanical resiliency.
5. The abstract should contain no references. Please modify based on the format requirement.
6. Is there any biofouling issue when applied to blood?
7. The authors claim that the supercapacitor performance was tested under blood flow in a PDMS microfluidic channel. How do they measure it? Do they fix the device or free the device in the flowing blood? Any mechanical force (e.g. crashing the channel walls) can influence the performance?
8. For the blood application, the authors should consider the biodegradability of the device and fate in the systemic circulation.
9. What's the obstacles to moving forward to the in vivo trial? How to take out the device from the blood after use? Please specify the difficulties.
10. Conclusions: can you elaborate more on the plan to translate this BSC system towards practical biomedical applications?

Reviewer #3:

Remarks to the Author:

The authors introduced an ultra-miniaturized electrochemical capacitor (Biosupercapacitor) fabricated in a swiss-roll geometry with a size of 0.001 mm³. This biosupercapacitor utilized biofluids such as blood as electrolyte for electrochemical energy storage. Furthermore, the authors constructed a pH sensor based on the change of the capacitance of their biosupercapacitor as a function of pH. The design of the biosupercapacitor of this work is interesting in terms of the size which allows implantation in the body's tiny spaces including blood vessels, and the manufacturing approach which allows good reproducibility. This work shed light on the challenges of the future power source of the miniaturized implantable bioelectronics and introduces tiny biosupercapacitors as a candidate alongside bio-batteries and biofuel cells. The manuscript is well-written and easy to follow and I believe the interdisciplinary nature of the manuscript is suitable for the diverse nature of the readers of Nat Comm. Although the manuscript provides useful insights as a step forward in the area of miniaturized implantable power sources, this work still suffers from a number of conceptual limitations (below) which render this work in it's current form not suitable as a real practical solution or as a

potential commercial device. Because of the novelty of this work and the insights it provides, I suggest considering this work for publications in Nat Comm if the major revisions provided below are carefully addressed.

Major comments:

1- The first conceptual limitation of this manuscript stems from the lack of device encapsulation:

A- The authors reported that the blood components such as enzymes and cells boost the capacitive behavior by 40% due to bioelectrocatalysis at the surface of the current collector which is reasonable. However, the manuscript fails to realize the impact of oxidizing and reducing the components of the blood during thousands of charge-discharge cycles as part of the biosupercapacitor operations. During this process all redox based components of the blood such as redox enzymes, GSH/GSSG system, and many others will be significantly disrupted. The authors could visualize the damaging effects by applying the same 5000 charge-discharge cycles of their device to a "redox dye" and observe the color changes. I suggest the authors to run biochemical assays of the blood redox systems before and after 5000-100,000 charge-discharge cycles and discuss this effect in the manuscript to highlight the limitation of this device for a long-term use.

B- Lack of encapsulation will necessarily lead to biofouling problems when the device is implanted for a period of days to weeks. The adsorption nature of many the whole blood components on metal surfaces (current collector) which is further maximized by the depositions of blood components due to charge-discharge cycles will eventually block the hollow tube of the "swiss-roll" device and compromise its performance. Biofouling and its impact should be assessed and discussed in the manuscript.

C- The authors charged the biosupercapacitor up to 1.6 V while blood is in contact with the current collector. This voltage will lead to a degree of electrochemical water splitting (90% of human blood is water). Please review OER and HER reactions of gold (current collector) in neutral media. It is important not charge the biosupercapacitor to a higher potential window than water splitting reactions to avoid damage. Please consider reducing the voltage, or justifying the safety of applying 1.6 V to blood.

2- Unlike batteries and fuel cells, the reported biosupercapacitor can not work alone and requires momentary charging source for continuous charging so that the biosupercapacitor can be practical. The authors in this work charged the biosupercapacitor from an outside source to use it as a power source for pH sensing. The authors need to discuss in their manuscript a real-life scenario of the source of power to charge such a biosupercapacitor to realize a functional power source for implantable device. Would this intended to be hybrid with a nanogenerator, a biofuel cell, etc. This discussion should be added to introduction and other parts of the manuscript as relevant. This will give the reader a direction of how this biosupercapacitor will be used in real implantation scenario.

3- The authors monitored the pH of the blood as a function of the capacitance change with a reasonable sensitivity. However, the selectivity of the sensor may not hold. In other words, the pH sensing was tested in DI water-based electrolyte at different pH, however, many factors can affect the capacitance of the biosupercapacitor if the test is to be performed in a complex media such as the blood. The authors should provide the sensitivity of this sensor in blood or at least plasma and discuss the results in the manuscript.

Minor comments:

4- The word supercapacitor in essence is a commercial terminology and not a scientific one even though it is commonly used in publications. This "super" terminology was used to indicate that the device stores more energy. The correct scientific name of the device is electrochemical capacitor, in this case, bioelectrochemical capacitor. I suggest the author to mention the word biosupercapacitor may be once and refer to the device as bioelectrochemical capacitor elsewhere in the MS.

5- I suggest the author to add the exact length and diameter of their biosupercapacitor to the text.

6- For visualization purposes, I suggest the authors to take a real photo of the device on a fingertip, if feasible. Although the authors showed a photograph under the microscope, a real photo would be useful to imagine how tiny it is compared to a fingertip, if feasible.

7- The authors should cite and discuss relevant biosupercapacitor literature such as Mosa, I. M.; Pattammattel, A.; Kadimisetty, K.; Pande, P.; El-Kady, M. F.; Bishop, G. W.; Novak, M.; Kaner, R. B.; Basu, A. K.; Kumar, C. V.; Rusling, J. F., Ultrathin Graphene-Protein Supercapacitors for

Miniaturized Bioelectronics. *Advanced Energy Materials* 2017, 7 (17), 1700358.

Pankratov, D.; Shen, F.; Ortiz, R.; Toscano, M. D.; Thormann, E.; Zhang, J.; Gorton, L.; Chi, Q., Fuel-independent and membrane-less self-charging biosupercapacitor. *Chemical Communications* 2018, 54 (83), 11801-11804.

Response to Reviewers

Our response to Reviewer #1:

Reviewer #1 comments: *This manuscript describes the preparation and characterization of supercapacitor devices that are composed of alternating layers of PEDOT-PSS that are separated by a PVA proton exchange layer. The method of preparing the devices is novel, relying on the self-assembly of the device into what is aptly described as a “Swiss roll” arrangement. This allows a device to be made that possesses the shape of a tube, one that is sufficiently small to be able to fit into blood vessels. Detailed characterization of the devices in terms of their properties and performance is presented and described in the figures, text and SI. Of note is the ability of the device to retain good capacitance and coulombic efficiency for 5000 cycles (fig 1). The device also possesses good biocompatibility properties and as described (lines 225 et seq.) no significant issues with inflammation etc. The results obtained from the characterization of the devices are promising and are indicative of a device that could be deployed and used in measurements for clinical applications. Overall, the level of detail on the preparation and characterization of devices is of a very good quality. The devices are incorporated into a self-powered pH sensor, one that can measure changes in pH over the range 1-7 (fig 4e). This represents a novel and a significant advance in making measurements of this type.*

In summary the manuscript describes a novel device that has significant potential for applications in clinical measurements, specifically in blood. Its of interest to a broad readership such as that of this journal. Prior to recommending it for publication the questions above on the modus operandi of the device in blood need to be addressed.

Thank you very much for your positive evaluation and your efforts on our manuscript. Your comments are very helpful for improving the quality of this work. We have supplemented more data and discussion according to your suggestions. The revised parts in the main text and supporting information are marked in blue. Our response is presented in the following:

Question 1.1: One aspect that is unclear is how the current collectors are physically connected to the potentiostat? Details of this are not evident in the methods and need to be.

Response 1.1: To physically connect to the Potentiostat (μ -Autolab) we used a micro-probe station (cascade Microtech). A microscope was used to align the probes to the bond pad area and make contact. Finally, the micro-probes are connected to the Auto lab work station using BNC connectors (see Figure R1). This discussion has been added to the methods section.

Figure R1 | Electrochemical measurement setup. **a**, Schematic illustration of nBSC measurement setup for all electrochemical analysis. **b**, Optical image of nBSC integrated into a PDMS micro fluidic channel with micro probes connected to the bond pads for electrochemical analysis. **c**, Optical image of nBSC measurement setup for all electrochemical analysis.

Question 1.2: It's also unclear as to how the device is charged for use? The device clearly operates as a supercapacitor but e.g., in figure 4 c, are the capacitors charged prior to use?

Response 1.2: As seen in Figure R2a the onboard nBSC is connected to an external potentiostat using charging lines (1,2) and the probe station setup described in Response 1.1. The charging lines (1,2) are also connected to the power lines (4,5) using the onboard interconnect lines (3). The output line (7) is connected to the signal line and power line (5) to the ground of an oscilloscope using a microprobe station. After completing the setup, initially the onboard nBSCs are charged to 3V (charging mode Figure R2b and SI Figure 29) using an external potentiostat through the charging line (1,2). Once the onboard nBSC is set to 3V, the external potentiostat is disconnected by removing the microprobe contacts from the charging lines (1,2). Thus, it allows the nBSC to discharge across the pH sensor through the interconnect (3), therefore powering the pH sensor and enabling autarkic operation. During the discharge phase of the nBSC the system is in sensing mode (see Figure R2c and SI Figure 30). In this mode, the pH sensor is fed with different pH solution through a microfluidic channel and the corresponding output is measured through the output line (7). The pH sensor was operated in 3 stages:

State1: Discharged state (initial state of the device)

State2: Charging of the nBSCs at $1\mu\text{A}$ for ~ 30 sec

State3: Discharged of the nBSCs across the pH sensor (measurement phase)

Figure R2 | Measurement configuration of self-powered pH sensor. **a**, Optical image of nBSC integrated self-powered pH sensor (top) before rolling, and (bottom) after rolling with all its connection lines (1. External charging line (+Ve) connected to potentiostat; 2. External charging line (-Ve) connected to potentiostat; 3. nBSC – pH sensor interconnect; 4. pH sensor power line (+Ve) connected to nBSC; 5. pH sensor groundline connected to nBSC; 6. Sensor unit test line; 7. pH sensor output to oscilloscope). **b**, Configuration of pH sensor in charging mode (external source charges the nBSC). **c**, Configuration of the pH sensor in measurement mode (nBSC powers the pH sensor).

Question 1.3: *The properties of the devices in blood are enhanced in comparison with 0.9% saline, however the explanation for this enhancement is not clear. The text “Blood consists of enzymes and living cells that produce strong bioelectrocatalytic reactions)” line 107 is correct in itself but does not apply clearly here. The electrode layers in the devices are enclosed and only exposed at the edge (the side view of the swiss roll) as in fig 2h. Its not clear that bioelectrocatalytic reactions can occur as such reactions require an electrode surface that usually has to undergo modification to enable electrocatalytic reactions to occur (and by definition electrocatalytic implies electron transfer). Reactions such as those that utilise ADP/ATP do not involve electron transfer, typically this requires the use of NAD(P) cofactors and/or redox active enzymes. This aspect of the manuscript needs to be addressed and a more detailed explanation of the effect in blood needs to be provided before*

Response 1.3: Blood is a complex fluid that contains proteins, glucose, ATP, numerous redox enzymes and ions, which possibly play an important role in the formation of electro-double-layers leading to the enhancement of the performance. Considering the complexity of biological fluids, it would be extremely difficult and out of the scope of this work to exactly identify which of the above components of blood are contributing to the electrochemical enhancement.

To answer the second part of the questions: Despite the coverage by the current collectors, the PEDOT electrodes can well interact with redox enzymes and plasma due to the presence of the PVA separator. PVA is a porous hydrogel matrix that efficiently absorbs and sucks in the electrolyte and its components at the edge of the Swiss-roll nBSC device [1]. As the PEDOT electrode surface is in tight contact with the PVA separator the redox enzymes in the blood can interact with the entire surface of the PEDOT, thus leading to significant performance enhancement. In order to estimate the absorption of the PVA separator we prepared PVA-SU8 stripes on glass substrate as shown in Fig. R3 a-d (see supplementary note 1 for describing the detailed fabrication process). These PVA-SU8 stripes were used to perform transmittance spectroscopy before and after dropping the three different electrolytes (0.9% NaCl, blood plasma and blood) at one end of the PVA-SU8 stripes. After allowing the liquid to diffuse for a few minutes, measurements were performed at a distance of 1.5 mm away from the electrolyte drop edge (see Fig. R3d,e) which is way beyond the full length of the nBSC (~1 mm). As shown in Fig. R3 f-h, after the introduction of the respective electrolytes (for optical absorption properties of electrolytes, see e.g. 0.9% NaCl [2-3], blood plasma [4-6] and blood [7-8]) the transmittance spectra at the measurement point are significantly decreased. This decrease in transmittance before and after introduction of the electrolyte shows that the electrolyte and all its components (redox enzymes and other components) diffuse through the PVA hydrogel separator and can thus provide efficient interaction with the PEDOT electrode surface. The above discussion has been added to the supporting information.

Figure R3 | Ability of PVA separator to absorb electrolyte. **a-b** Schematic illustrating the fabrication of the PVA-SU8 stripe for the electrolyte absorption test. **(a)** top view and **(b)** side view. **c**, Microscope image of PVA-SU8 stripe for transmittance spectroscopy. **d**, (top) Magnified image of PVA-SU8 stripe showing individual components. (bottom) Electrolyte drop position and measurement point on PVA-SU8 stripe. **e**, (left) Optical image of PVA-SU8 separator stripe with blue dye-NaCl electrolyte, (center) PVA-SU8 separator stripe with blood plasma electrolyte and (right) PVA-SU8 separator stripe with blood electrolyte. **f**, Transmittance spectrum through PVA-SU8 stripe with absorbed blue dye-NaCl electrolyte. **g**, Transmittance spectrum through PVA-SU8 stripe with absorbed blood plasma electrolyte. **h**, Transmittance spectrum through PVA-SU8 stripe with absorbed blood electrolyte.

Question 1.4: An additional comment is that the manuscript would benefit from a discussion on the size of the devices in terms of the radius of the channel, ca 30 μm .

a. Was this radius selected or a consequence of the method of preparation?

b. Can the size be altered and if so over what range?

c. Does this size exclude cells (RBCs have a radius of ca 6 μm and should be able to enter the channel).

Response 1.4: After the rolling process, the device transforms into a Swiss-roll tube, 1mm in length and with an inner diameter of $\sim 25 \mu\text{m}$ (~ 3 times bigger than red blood cells) and an external diameter of $\sim 50 \mu\text{m}$. Due to several windings, the outer diameter is larger than the inner diameter. The particular number of windings and tube diameters were selected on purpose to achieve two goals. First, create an as-compact-as-possible device and second, ensure to have enough room for blood and all its components to flow freely through the tube. The outer diameter of the tubes – incorporating only the polymeric rolling stack layers – changes from 120 μm to about 40 μm (see Figure R4a) if the pH value of the rolling solution is changed from 6 to 10. With all nBSC layers included in the device, the diameter can be changed from $\sim 220 \mu\text{m}$ to $50 \mu\text{m}$ (see Figure R4b). The hydrogel used in the rolling stack is a pH trigger hydrogel. When the pH of the rolling solution increases the absorption capability of the hydrogel layer is enhanced leading to the generation of larger strain and hence a smaller tube diameter. This discussion has been added to the supporting information.

Figure R4 | Effect of rolling solution pH on nBSC diameter. **a**, (left) Top-view optical images of bare polymer tubes rolled in different pH solutions after 20h. (right) Tube diameter as a function of rolling solution pH and time of rolling. **b**, (left) Top-view optical images of “Swiss-roll” nBSCs in different pH solutions after 20h. (right) Tube diameter as a function of rolling solution pH and time of rolling. Scale bar, 150 μm (**a-b**). Error bars represent the variation of tube diameter over fifty measured devices. Dotted lines indicate the projected vertical walls of the tubes (**a**).

Our response to Reviewer #2:

Reviewer #2 comments: *In this interesting work, Lee et al. report on the development of a roll-up micron-sized biosupercapacitor with enzyme-aided reaction to combat the self-discharge issue commonly faced by miniaturized supercapacitors. To demonstrate the application of such energy storage device, the authors developed an integrated pH sensing system powered by such nano-biosupercapacitor. The work is novel and the study is well done and of high quality. However, the visionary concept proposed by the authors has some limitations towards practical biomedical application, which should be addressed and discussed.*

Thank you very much for your positive evaluation and your efforts on our manuscript. Your comments are very helpful for improving the quality of this work. We have supplemented more data and discussion according to your suggestions. The revised parts in the main text and supporting information are marked in blue. Our response is presented in the following:

Question 2.1: *As we know, the stability window in aqueous environment is only 1.21 V. The operation of the supercapacitor at 3 V is out of the stability window of the blood and may cause gas evolution. How is the device addressing this issue? Furthermore, as the BSC requires direct contact with the blood to maintain the voltage, how can three BSCs connect in series while sharing the same electrolyte?*

Response 2.1: We agree with the reviewer that the water splitting can occur at 1.2V, immediately raising concerns about how the device can operate at 3V without any significant gas evolution or water splitting. To suppress water splitting and enable stable operation, nBSCs were passivated with a 500 nm thick insulating SU8 photoresist layer isolating the system electrically and ionically. This quasi-electronic and ionic isolation significantly reduces the gas evolution and water-splitting reaction. As seen in Figure R5a and R5b, the devices with SU8 photoresist passivation (before and after rolling) show no signs of degradation, gas evaluation or water splitting. In contrast, the non-passivated devices show significant degradation when charged to 1.6 V and significant gas evolution due to water splitting is observed (see Figure R6a and Supplementary video 1). Moreover, the non-passivated devices were also damaged during the rolling process yielding a non-functional device (see Figure R6b). The above discussion has been added to the supporting information. And the supporting information Figure 1 has been modified (SU8 photoresist passivation layer was added).

Figure R5 | Effect of SU8 photoresist passivation on device and electrolyte stability. a, Electrolyte stability of SU8 photoresist passivated device. (left) optical images of stable SU8 photoresist passivated device after charging to different voltages. (right) GCD curves at four different potential windows at 500 nA. **b,** Electrolyte stability of “Swiss-roll” SU8 photoresist passivated device. (left) optical images of stable “Swiss-roll” SU8 photoresist passivated device after charging to different voltages. (right) GCD curves at four different potential windows at 500 nA. Scale bar 150 μm (a-b).

Figure R6 | Effect of SU8 photoresist passivation on device and electrolyte stability. a, Electrolyte stability of non-passivated device. (left) optical images of unstable non-passivated device after charging to different voltages. (right) GCD curves at four different potential windows at 500 nA. **b,** optical images of broken “Swiss-roll” non-passivated device. Scale bar 150 μm (a-b).

Regarding the second part of the question: The three BSCs connected in series can charge up to 3V while maintaining contact with the blood due to the quasi-electrical isolation provided by the SU8 photoresist passivation and the PVA hydrogel separator. The SU8 photoresist passivates the complete nBSC except for a small portion of the electrically neutral PVA hydrogel separator. The SU8 photoresist passivation includes the current collectors, electrodes, and interconnects (as indicated by the blue dotted line in Fig. R7a) protecting the nBSC from all possible electrical short-circuit paths through the electrolyte. Additionally, the electrochemical impedance of the PVA hydrogel separator at the edge of the nBSC tube is extremely high. This property avoids any ionic short-circuit paths while the PVA absorbs and supplies the electrolyte to the PEDOT electrode surface for electrochemical charge storage. In order to prove this, we prepared three nBSCs connected in series with and without SU8 photoresist passivation as shown in Fig R7a-b. These series-connected nBSCs were charged to 3V with 100 nA. As shown in Fig R7c-d, the nBSC devices which are passivated reach the charging voltage of 3V. In contrast, the non-passivated device could only reach a maximum single cell potential

of 1.5V at the same charging current showing that electronic and ionic isolated SU8 photoresist devices can be connected in series and charge to high voltage while sharing the same electrolyte.

Figure R7 | Effect of SU8 photoresist passivation on device and electrolyte stability. **a**, Microscope image of three nBSCs connected in series and passivated with SU8 photoresist. **b**, Microscope image of three nBSCs connected in series without SU8 photoresist passivation (scale bar, 150 μ m; blue dotted line shows SU8 photoresist passivation area and the white dotted line shows PVA separator area). **c**, GCD curve of SU8 photoresist passivated device charged to 1, 2 and 3V at 100 nA. **d**, GCD curve of non-passivated device charged to 1 and 1.5V at 100 nA.

Supplementary Figure 1 | Schematic illustration of nBSC fabrication. **a**, Strained polymeric rolling. **b**, Bottom current collector, Cr/Au. **c**, Bottom electrode, PEDOT: PSS. **d**, Proton exchange separator, photo-patterned PVA. **e**, Top electrode, PEDOT: PSS. **f**, Top current collector, Cr/Au. **g**, Completed flat nBSC with SU8 photoresist passivation. **h**, Etching and rolling procedure. **i**, Final rolled-up nBSC.

Question 2.2: *The roll-up device occupies a significant amount of volume in the blood vessel, and can cause a significant increase in local blood pressure and may cause a reduction in flow rate. The authors propose to insert the device inside the blood vessels. Some comments on this concern would be helpful*

Response 2.2: The nBSC device occupies a volume of ~1 to 2 nl (or ~1E-3 to 2E-3 mm³) which is significantly smaller than many blood vessels (e.g. arteries or veins). Moreover, the hollow core of the tube provides a free-flow channel for blood to pass. Blood vessels have different diameters at various

parts of the body, so to understand the blood flow profile through a vessel we have performed flow simulations of a nBSC placed at the wall of the blood vessel (see Figure R8, R9, R10 and Supplementary video 7-8). As seen in the simulation videos, the nBSC placed at the wall would minimally increase the local blood pressure ($\pm 10\%$). This change in pressure is correlated to the average change of velocity $\pm 30\%$. We have supplemented additional experiments (see Figure R11 and Supplementary video 3-6) by placing the nBSC in a two-phase flow microfluidic channel to validate the simulations. Dulbecco's buffered saline (DPBS) was constantly pumped through two microfluidic channel inlets at a flow rate of 1 mm/s and additionally, blood was injected through the third channel inlet (see Figure R11). This setup allows the visualization of the flow around the nBSC. As shown in the Supplementary videos 3-6, the flow profile of blood correlates with the simulations for the nBSC placed at the wall of the vessel.

In order for the nBSC to not cause any significant health risks the nBSC should not block the blood vessel by more than 30 % [9]. Thus, by choosing a sufficiently large blood vessel diameter or by tuning the inner/outer diameter of the nBSC, one can implant the device without causing major vascular blockage ($< 30\%$) or significant health risks. The above discussion has been added to the supporting information.

Figure R8 | Simulation of blood pressure and flow velocity profile as a function of nBSC outer diameter. **a**, Pressure profile of the blood vessel as a function of nBSC outer diameter changing from 50 to 250 μm . **b**, Pressure profile of the blood vessel as a function of nBSC inner diameter changing from 10 to 240 μm (which accounts to 96 % to 4 % blockage of blood vessel). **c**, Velocity profile of the blood vessel as a function of nBSC outer diameter changing from 50 to 250 μm . **d**, Velocity profile of the blood vessel as a function of nBSC inner diameter changing from 10 to 240 μm .

Figure R9 | Simulation of blood pressure and flow velocity profile as a function of nBSC outer diameter. **a-b,** Effect of nBSC outer diameter on local blood pressure. **a,** Relative systolic pressure changes along the length of blood vessel as a function of nBSC outer diameter. **b,** Relative diastolic pressure changes along the length of blood vessel as a function of nBSC outer diameter. **c-d,** Effect of nBSC diameter on local blood velocity. **c,** Relative systolic velocity change along the length of blood vessel as a function of nBSC outer diameter. **d,** Relative diastolic velocity change along the length of blood vessel as a function of nBSC outer diameter.

Figure R10 | Simulation of blood flow velocity profile as a function of nBSC inner diameter. a-b, Effect of nBSC inner diameter on local blood pressure. **a,** Relative systolic pressure changes along the length of blood vessel as a function of nBSC inner diameter. **b,** Relative diastolic pressure changes along the length of blood vessel as a function of nBSC inner diameter. **c-d,** Effect of nBSC inner diameter on local blood velocity. **c,** Relative systolic velocity change along the length of blood vessel as a function of nBSC inner diameter. **d,** Relative diastolic velocity change along the length of blood vessel as a function of nBSC inner diameter.

Figure R11 | Experimental verification: Impact of nBSC on blood flow profile. a, Optical image of two-phase flow microfluidic channel measurement setup with phase1: DPBS; Phase2: Blood

Question 2.3: *How will the rolled-up BSC be charged while implanted within the blood vessel? Is it through a wired connection? If so, how to connect to the BSC through the blood vessel?*

Response 2.3: Finding an efficient way to charge the nBSCs tetherlessly is a crucial future goal. Ideally, implanted nBSCs should be charged either wirelessly or via integrated energy generators (such as: piezoelectric, triboelectric or pyroelectric generators). As suggested by reviewer #3 it would be interesting to couple the nBSC with a generator or biofuel cell to form a hybrid energy source. Currently, the nBSC can be powered and charged through external cables. But for real-life applications, it is more promising to for instance integrate the device with a “Swiss-roll” helical micro-coil. The coil can be used for wireless power transfer via RF or inductive-coupling. “Swiss-roll” helical micro-coils for medical implant applications have been reported, previously [10]. The above discussion on potential wireless power transfer for *in-vivo* charging of nBSCs has been added to the conclusion section.

Question 2.4: *The blood vessel dilates and contracts for tens to hundreds of times per minute, which might cause repeated mechanical deformations on the BSC device. Figure 3h have shown the stability of the BSC in 75 cycles of compression, but its stability through more deformation cycles are needed to ensure its mechanical resiliency.*

Response 2.4: Performance stability of energy storage as well as microelectronic devices are affected by mechanical deformation and loads [11-13]. A common way to estimate the lifetime of these devices is to apply stress or pressure several times higher than encountered by the devices under normal conditions. Such methods are referred to as dynamic stress testing [11] and accelerated lifetime testing [14-15]. Due to the non-linear failure mechanisms, the inferred lifetime of the devices under normal operational conditions would be significantly higher than that experienced under the extreme conditions during lifetime testing experiments [16]. While we feel that rigorous and exhaustive lifetime testing is beyond the scope of this work, we were still interested in the mechanical resiliency of nBSCs and carried out a few additional experiments at 5 kPa and 15 kPa – the latter being three times higher than the normal hydrostatic pressures (2.0 to 5.3 kPa) encountered by the devices. As shown in Figure 3h and R12a, the nBSC can withstand a pressure of 15 kPa showing a capacitance retention of nearly ~100 % over 100 compression cycles. We also performed repeated compressions at normal hydrostatic pressure of 5 kPa. As shown in Figure R12b, the nBSC also shows stable performance with a capacitance retention above 80 % over 200 cycles.

Figure R12 | Compression measurements of nBSCs. **a**, Capacitance retention and coulombic efficiency of a nBSC under repeated compressions at 15 kPa over 75 cycles. Inset shows 75x GCD cycles (each GCD curve was measured after 5 compression cycles) of nBSC subjected to 15 kPa. **b**, Capacitance retention and coulombic efficiency of a nBSC under repeated compressions at 5 kPa over 200 cycles. Inset shows 200x GCD cycles (each GCD curve was measured after 20 compression cycles) of nBSC subjected to 5 kPa. (all measurements were performed at 25 °C and under static conditions (0 ml/min).)

Question 2.5: *The abstract should contain no references. Please modify based on the format requirement.*

Response 2.5: The abstract has been adapted to Nat. Comm. format.

Question 2.6: *Is there any biofouling issue when applied to blood?*

Response 2.6: In the flow test videos 3-6 and Figure R11 biofouling was not observed around the nBSC devices. Admittedly, the maximum amount of time the device was operated under dynamic flow conditions was only 5.5 days. Biofouling mainly refers to protein adsorption, thus coatings which can

heparinize the surface to control and reduce fibrin deposition are desirable. Many anti-fouling strategies have been investigated and studied over the years to prevent fouling in vascular stents and catheters [17-21] providing promising options to prevent biofouling in our nBSCs. One possible anti-fouling strategy is to use suitable materials for encapsulation and surface-repellent coatings. Polyethylene glycol (PEG), also known as polyethylene oxide (PEO), is highly hydrophilic and prevents protein and cellular adsorption [22]. Another example is coating with pyrolytic carbon which is used in a variety of vascular settings including heart valves [17]. Pyrolytic carbon offers good biocompatibility and demonstrates decreased platelet adhesion when compared to other materials [13]. To avoid biofouling due to bacteria simple silver coating is another option [24]. All these materials can be incorporated into the fabrication protocol of the nBSC device thus providing a good way to tackle the biofouling issue.

Question 2.7: *The authors claim that the supercapacitor performance was tested under blood flow in a PDMS microfluidic channel. How do they measure it? Do they fix the device or free the device in the flowing blood? Any mechanical force (e.g. crashing the channel walls) can influence the performance?*

Response 2.7: Figure R13 illustrates the preparation and integration of the nBSC into a PDMS microfluidic channel. The nBSC devices are fixed on a wafer and integrated with the microfluidic channel. For measurements, the nBSCs were physically connected to the Potentiostat (μ -Autolab) using a microprobe station (cascade Microtech). As shown in Fig. R13, the probe station micro-probes are connected to the device bond pads. A microscope was used to align the probes to the bond pad area and make contact. Finally, the micro-probes are connected to the Auto-lab work station using BNC connectors. The measurement set-up is very stable and was used to study the influence of changing the electrolyte on the device electrochemical performance without any mechanical reconfigurations. The discussion has been added to the method section of the manuscript. Moreover, to address the last part of the question, as shown in Fig. 3g-h and SI Fig. 24, the nBSC can withstand mechanical deformations up to a pressure of 60 kPa.

Figure R13 | Schematic illustration of nBSC fabrication and integration into a microfluidic channel. a, 250 μ m thick spin coated SU8 photoresist layer on Si/SiO₂. **b,** Patterned microfluidic channel mold with UV exposure. **c,** PDMS casting on the micro fluidic SU8 photoresist mold. **d,** PDMS curing around SU8 photoresist mold. **e,** Removal of PDMS microfluidic channel from the mold. **f,** Surface actuation of nBSC chip and PDMS microfluidic channel. **g,** Mechanical bonding of PDMS microfluidic channel with nBSC chip. **h,** Complete integration of micro fluidic channel with nBSC. **i,** Schematic illustration of micro fluidic channel integrated nBSC with fluid inlet and outlet and physical contact to measurement platform via micro probes.

Question 2.8: For the blood application, the authors should consider the biodegradability of the device and fate in the systemic circulation.

Response 2.8: At the current state, the materials used in the nBSC are not biodegradable. This is why we performed the cell viability test (cytotoxic test) and clotting and inflammation test (chemotoxicity) of different materials and a complete device. As presented in SI Fig. 25-26, the devices showed low to moderate cell viability and inflammation. This leads us to conclude that the device would not have a significant toxicity issue for the body. Furthermore, such non-biodegradable nBSCs could be interesting for acute biomedical applications such as smart catheters and biomedical sensors. But we agree with the reviewer that one should also focus on biodegradable nBSCs in future studies. However, as this is not the main focus of the work, biodegradable nBSCs are discussed in the conclusion section as a future perspective.

Question 2.9: What's the obstacles to moving forward to the *in vivo* trial? How to take out the device from the blood after use? Please specify the difficulties.

Response 2.9:

A few obstacles to move forward for *in-vivo* applications are:

1. Remote wireless charging (addressed above).

2. Microsystem integration of nBSCs with ultra-compact electronics and multifunctional but selective sensor layouts
2. Sufficient power density and energy density for long time usage.
4. Active biodegradability of the device upon failure (addressed above).
5. Possible immunoreactions, which can be overcome by employing cell-membrane camouflages or coatings to prolong life time of device.

The device can be taken out by minimally invasive surgery or better, by active biodegradability as outlined above.

***Question 2.10:** Conclusions: can you elaborate more on the plan to translate this BSC system towards practical biomedical applications?*

Response 2.10: In order to translate the nBSC devices into real biomedical applications several challenges need to be addressed: The ability to charge the nBSCs *in-vivo* without tether; device encapsulation against immunoreactions and biofouling caused by proteins, cells, bacteria and others; a smart way to implant the nBSCs in a vascular network with minimum invasion; possible biodegradation of non-functional devices. Overcoming these challenges by the materials versatility, the design flexibility as well as the extreme integration capabilities of on-chip manufacturing render these devices particularly promising microsystems for self-powered biomedical implants, smart cardiovascular stents and autarkic motile microelectronic systems.

Our response to Reviewer #3:

Reviewer #3 comments: *The authors introduced an ultra-miniaturized electrochemical capacitor (Biosupercapacitor) fabricated in a swiss-roll geometry with a size of 0.001 mm³. This biosupercapacitor utilized biofluids such as blood as electrolyte for electrochemical energy storage. Furthermore, the authors constructed a pH sensor based on the change of the capacitance of their biosupercapacitor as a function of pH. The design of the biosupercapacitor of this work is interesting in terms of the size which allows implantation in the body's tiny spaces including blood vessels, and the manufacturing approach which allows good reproducibility. This work shed light on the challenges of the future power source of the miniaturized implantable bioelectronics and introduces tiny biosupercapacitors as a candidate alongside bio-batteries and biofuel cells. The manuscript is well-written and easy to follow and I believe the interdisciplinary nature of the manuscript is suitable for the diverse nature of the readers of Nat Comm. Although the manuscript provides useful insights as a step forward in the area of miniaturized implantable power sources, this work still suffers from a number of conceptual limitations (below) which render this work in it's current form not suitable as a real practical solution or as a potential commercial device. Because of the novelty of this work and the insights it provides, I suggest considering this work for publications in Nat Comm if the major revisions provided below are carefully addressed.*

Thank you very much for your positive evaluation and your efforts on our manuscript. Your comments are very helpful for improving the quality of this work. We have supplemented more data and discussion according to your suggestions. The revised parts in the main text and supporting information are marked in blue. Our response is presented in the following:

Question 3.1a: 1- *The first conceptual limitation of this manuscript stems from the lack of device encapsulation:*

The authors reported that the blood components such as enzymes and cells boost the capacitive behavior by 40% due to bioelectrocatalysis at the surface of the current collector which is reasonable. However, the manuscript fails to realize the impact of oxidizing and reducing the components of the blood during thousands of charge-discharge cycles as part of the biosupercapacitor operations. During this process all redox based components of the blood such as redox enzymes, GSH/GSSG system, and many others will be significantly disrupted. The authors could visualize the damaging effects by applying the same 5000 charge-discharge cycles of their device to a “redox dye” and observe the color changes. I suggest the authors to run biochemical assays of the blood redox systems before and after 5000-100,000 charge-discharge cycles and discuss this effect in the manuscript to highlight the limitation of this device for a long-term use.

Response 3.1a: The human body typically accommodates 5 liters of blood, which constantly circulates through the body [25-26]. This blood is reused by the body several times and undergoes constant replenishment through efficient recycling and detoxification processes present in our kidney. The introduction of a nBSC into a blood vessel would not significantly consume redox enzymes and other chemicals from the blood. A simple estimation tells us that the hydrogel layer and the PVA separator in the nBSC absorb approximately ~0.5 nl of electrolyte during a full blood circulation. This volume is 10 billion times smaller than the total blood volume in an average human body and would not cause any health risk [27-30]. Nevertheless, in order to check out any change in chemical composition of biological electrolytic fluid we fabricated nBSCs integrated into a PDMS microfluidic channel and prepared 5 ml of two different electrolyte systems: pure blood plasma and blood plasma mixed with 0.5 % methyl viologen dichloride hydrate (redox dye). The redox dye was introduced to blood plasma in order to visualize the active reduction of electrolyte during the charging phase of the nBSC. After the electrolyte preparation, 400 μ l of both electrolytes were separately collected as a reference for absorption spectrophotometric characterization. Following this, the microfluidic channel was connected to a syringe pump generating a forth-and-back flow rate of 100 μ l min^{-1} of the remaining blood plasma redox electrolytes. We then performed 5000x galvanostatic charge-discharge (GCD) cycling of the nBSC device (see Figure R14a) and finally a 400 μ l sample of the electrolyte was collected at the end of the 5000x cycles for absorption spectrophotometric characterization. As compared in Figure R14b and Supplementary video 2, the redox dye blood plasma does not show any

evident change in absorption spectrum or color of the blood plasma electrolyte during charging or discharging. In order to confirm this, we also performed 5000x GCD cycles on the pure blood plasma system as a control. As shown in Figure R14c-d, the absorption spectrum of the blood plasma collected after 5000x GCD cycles does not show any significant deviation compared to the reference before cycling, either. This result combined with the large volume of blood and natural recycling system present in an average human body lead us to conclude that the nBSCs would not pose any health risks as there are no significant changes to the chemical compositions of these fluids.

Figure R14 | Effect of multiple GCD cycling on blood plasma chemical compositions. **a**, Capacitance retention and coulombic efficiency of a device in blood plasma including 0.5% redox dye over 5000x GCD cycles. Inset shows GCD curves at 50 nA. **b**, Spectrophotometric characterization of blood plasma treated with 0.5% redox dye before and after 5000x GCD cycles. **c**, Capacitance retention and coulombic efficiency of a device in blood plasma over 5000x GCD cycles. Inset shows GCD curves at 50 nA. **d**, Spectrophotometric characterization of blood plasma before and after 5000x GCD cycles.

Question 3.1b: *Lack of encapsulation will necessarily lead to biofouling problems when the device is implanted for a period of days to weeks. The adsorption nature of many the whole blood components on metal surfaces (current collector) which is further maximized by the depositions of blood components due to charge-discharge cycles will eventually block the hollow tube of the “swiss-roll” device and compromise it’s performance. Biofouling and it’s impact should be assessed and discussed in the manuscript.*

Response 3.1b: In the flow test videos 3-6 and Figure R11 biofouling was not observed around the nBSC devices. Admittedly, the maximum amount of time the device was operated under dynamic flow conditions was only 5.5 days. Biofouling mainly refers to protein adsorption, thus coatings which can heparinize the surface to control and reduce fibrin deposition are desirable. Many anti-fouling strategies have been investigated and studied over the years to prevent fouling in vascular stents and catheters [17-21] providing promising options to prevent biofouling in our nBSCs. One possible anti-fouling strategy is to use suitable materials for encapsulation and surface-repellent coatings. Polyethylene glycol (PEG), also known as polyethylene oxide (PEO), is highly hydrophilic and prevents protein and cellular adsorption [22]. Another example is coating with pyrolytic carbon which is used in a variety of vascular settings including heart valves [17]. Pyrolytic carbon offers good biocompatibility and demonstrates decreased platelet adhesion when compared to other materials [23]. To avoid biofouling due to bacteria simple silver coating is another option [24]. All these materials can be incorporated into the fabrication protocol of the nBSC device thus providing a good way to tackle the biofouling issue.

Question 3.1c: The authors charged the biosupercapacitor up to 1.6 V while blood is in contact with the current collector. This voltage will lead to a degree of electrochemical water splitting (90% of human blood is water). Please review OER and HER reactions of gold (current collector) in neutral media. It is important not charge the biosupercapacitor to a higher potential window than water splitting reactions to avoid damage. Please consider reducing the voltage, or justifying the safety of applying 1.6 V to blood.

Response 3.1c:

We agree with the reviewer that the water splitting can occur at 1.2V, immediately raising concerns about how the device can operate at 3V without any significant gas evolution or water splitting. To suppress water splitting and enable stable operation nBSCs were passivated with a 500 nm thick insulating SU8 photoresist layer isolating the system electrically and ionically. This quasi-electronic and ionic isolation significantly reduces the gas evolution and water-splitting reaction. As seen in Figure R5a and R5b, the devices with SU8 photoresist passivation (before and after rolling) show no signs of degradation, gas evaluation or water splitting. In contrast, the non-passivated devices show significant degradation when charged to 1.6 V and significant gas evolution due to water splitting is observed (see Figure R6a and Supplementary video 1). Moreover, the non-passivated devices were

also damaged during the rolling process yielding a non-functional device (see Figure R6b). The above discussion has been added to the supporting information. And the supporting information Figure 1 has been modified (SU8 photoresist passivation layer was added).

Question 3.2: *Unlike batteries and fuel cells, the reported biosupercapacitor can not work alone and requires momentary charging source for continuous charging so that the biosupercapacitor can be practical. The authors in this work charged the biosupercapacitor from an outside source to use it as a power source for pH sensing. The authors need to discuss in their manuscript a real-life scenario of the source of power to charge such a biosupercapacitor to realize a functional power source for implantable device. Would this intended to be hybrid with a nanogenerator, a biofuel cell, etc. This discussion should be added to introduction and other parts of the manuscript as relevant. This will give the reader a direction of how third biosupercapacitor will be used in real implantation scenario.*

Response 3.2: Finding an efficient way to charge the nBSCs tetherlessly is a crucial future goal. Ideally, implanted nBSCs should be charged either wirelessly or via integrated energy generators (such as: piezoelectric, triboelectric or pyroelectric generators). As suggested by reviewer #3, it would be interesting to couple the nBSC with a generator or biofuel cell to form a hybrid energy source. Currently, the nBSC can be powered and charged through external cables. But for real-life applications it is more promising to for instance integrate the device with a “Swiss-roll” helical micro-coil. The coil can be used for wireless power transfer via RF or inductive-coupling. “Swiss-roll” helical micro-coils for medical implant applications have been reported, previously [10]. The above discussion on potential wireless power transfer for *in-vivo* charging of nBSCs has been added to the conclusion section.

Question 3.3: *The authors monitored the pH of the blood as a function of the capacitance change with a reasonable sensitivity. However, the selectivity of the sensor may not hold. In other words, the pH sensing was tested in DI water-based electrolyte at different pH, however, many factors can affect the capacitance of the biosupercapacitor if the test is to be performed in a complex media such as the blood. The authors should provide the sensitivity of this sensor in blood or at least plasma and discuss the results in the manuscript.*

Response 3.3: We agree with the reviewer. Complex biological fluids like blood indeed have several other components that would influence the pH sensor's selectivity and sensitivity. As requested by the

reviewer, we performed additional experiments by subjecting the pH sensor into blood plasma. We altered the pH of blood plasma by adding and adjusting the concentration of citric acid to achieve a blood plasma pH from 1 to 7. The self-powered pH sensor was used to sense the pH of the altered blood plasma (Figure R15a). The sensor showed a relative frequency change of $-2.7 \text{ E-}2 \pm 2.0 \text{ E-}2$ per pH and yields larger error bars in blood plasma compared to the relative frequency change of $-2.4 \text{ E-}2 \pm 1.9 \text{ E-}4$ per pH in artificial plasma (Figure R15b). The larger error bar reflects the complex activity of the enzymes present in blood plasma. This is due to the relative broadening of the pH spectrum (Figure R15b inset) in blood plasma compared to artificial plasma (Figure 4d). As observed in the frequency spectral response of the nBSC (Figure R15b inset), broader and several overlapping frequency peaks are typical for the response in blood plasma compared to the narrow frequency spectral response of the nBSC in artificial plasma (Figure 4d). This indicates that the complex composition of the biological fluid consisting of glucose, redox enzymes, and other chemicals influences the capacitance of the nBSC and affects the sensitivity. The above discussion has been added to the supporting information.

Figure R15 | nBSC as a self-powered pH sensor. a, Optical image showing all components of self-powered nBSC based pH sensor integrated in microfluidic channel in electrolyte flow. **b,** Relative frequency change of nBSC as a function of electrolyte pH of biological and artificial plasma. Inset figure: Frequency spectral response of the nBSC based pH sensor in pH 3 blood plasma.

Question 3.4: *The word supercapacitor in essence is a commercial terminology and not a scientific one even though it is commonly used in publications. This “super” terminology was used to indicate that the device stores more energy. The correct scientific name of the device is electrochemical capacitor, in this case, bioelectrochemical capacitor. I suggest the author to mention the word biosupercapacitor may be once and refer to the device as bioelectrochemical capacitor elsewhere in the MS.*

Response 3.4: We agree with the reviewer that the word “supercapacitor” is a more commercial terminology than a scientific one. However, many publications use the term “supercapacitor” (including the references suggested by the reviewer), and because the term “supercapacitor” resonates with a broader audience, we would like to use it instead of “electrochemical capacitor”. In order to keep the scientific terminology accurate, we have modified the introduction section and now write ‘The most commonly available energy storage units at the submillimeter scale are micro-electrochemical capacitors also termed micro-supercapacitors’.

Question 3.5: I suggest the author to add the exact length and diameter of their biosupercapacitor to the text.

Response 3.5: The nBSC has the following dimensions:

Samples in the planar form are 1 mm in length and have a width (rolling distance) of ~0.5 mm. After the rolling process, the device transforms into a Swiss-roll tube, 1mm in length and with an inner diameter of ~25 μm . Due to several windings the outer diameter is larger (~50 μm) than the inner diameter. This discussion has been added to the main text.

Question 3.6: For visualization purposes, I suggest the authors to take a real photo of the device on a fingertip, if feasible. Although the authors showed a photograph under the microscope, a real photo would be useful to imagine how tiny it is compared to a fingertip, if feasible

Response 3.6:

Figure R16 | Array of 90 nBSCs on fingertip.

Comment 3.7: *The authors should cite and discuss relevant biosupercapacitor literature such as*

Mosa, I. M.; Pattammattel, A.; Kadimisetty, K.; Pande, P.; El-Kady, M. F.; Bishop, G. W.; Novak, M.; Kaner, R. B.; Basu, A. K.; Kumar, C. V.; Rusling, J. F., Ultrathin Graphene–Protein Supercapacitors for Miniaturized Bioelectronics. Advanced Energy Materials 2017, 7 (17), 1700358. Pankratov, D.; Shen, F.; Ortiz, R.; Toscano, M. D.; Thormann, E.; Zhang, J.; Gorton, L.; Chi, Q., Fuel-independent and membrane-less self-charging biosupercapacitor. Chemical Communications 2018, 54 (83), 11801-11804.

Response 3.7: References have been added at appropriate locations.

References

1. Jensen, B. E. B, Dávila, I. & Zelikin, A. N. Poly(vinyl alcohol) physical hydrogels: matrix-mediated drug delivery using spontaneously eroding substrate. *J Phys Chem B* **26**, 120 (2016).
2. Ahmed, A. A. A., Al-Hussam, A. M., Abdulwahab, A. M. & Ahmed, A. N. A. A. The impact of sodium chloride as dopant on optical and electrical properties of polyvinyl alcohol. *AIMS Mater Sci* **5**, 533 (2018).
3. Sanyal, S., Bhui, U. K., Kumar, S. S. & Balaga, D. Designing injection water for enhancing oil recovery from kaolinite laden hydrocarbon reservoirs: A spectroscopic approach for understanding molecular level interaction during saline water flooding. *Energy Fuels* **31**, 11627 (2017).
4. Pang, J. *et al.* Green aqueous biphasic systems containing deep eutectic solvents and sodium salts for the extraction of protein. *RSC Adv* **7**, 49361 (2017).
5. Hu, Y. J., Yang, Y. O. & Zhang, Y. Affinity and specificity of ciprofloxacin-bovine serum albumin interactions: Spectroscopic approach. *Protein J* **29**, 234 (2010).
6. Shilova, O. N., Shilov, E. S. & Deyev, S. M. The effect of trypan blue treatment on autofluorescence of fixed cells. *Cytom A* **91**, 917 (2017).
7. Tuchin, V. V., Zhestkov, D. M., Bashkatov, A. N. & Genina, E. A. Theoretical study of immersion optical clearing of blood in vessels at local hemolysis. *Opt Express* **12**, 2966 (2004).
8. Mallya, M. *et al.* Absorption spectroscopy for the estimation of glycated hemoglobin (HbA1c) for the diagnosis and management of diabetes mellitus: A pilot study. *Photomed Laser Surg* **31**, 219 (2013).
9. McMains, V. & Nelson, L. Updated classification system captures many more people at risk for heart attack. Available from: https://www.hopkinsmedicine.org/news/media/releases/updated_classification_system_captures_many_more_people_at_risk_for_heart_attack (2017).
10. Karnaushenko, D. D., Karnaushenko, D., Makarov, D. & Schmidt, O. G. Compact helical antenna for smart implant applications. *NPG Asia Mater* **7**, 188 (2015).
11. Lin, Y. C. & Chung, K. J. Lifetime prognosis of lithium-ion batteries through novel accelerated degradation measurements and a combined gamma process and monte carlo method. *Appl Sci* **9**, 559 (2019).
12. Xu, R. & Zhao, K. Electrochemomechanics of electrodes in Li-ion batteries: A review. *J Electrochem Energy* **13**, 030803 (2016).
13. Bucci, G. *et al.* The effect of stress on battery-electrode capacity. *J Electrochem Soc* **164**, 645 (2017).

14. Nelson, W. Accelerated life testing-step-stress models and data analyses. *IEEE Trans Reliab* **29**, 103 (1980).
15. Spencer, F. W. Statistical methods in accelerated life testing. *Technometrics* **33**, 360 (1991).
16. Pham, H. Handbook of reliability engineering. Springer. ISBN 1-85233-453-3 (2003).
17. Wallace, A. et al. Anti-fouling strategies for central venous catheters. *Cardiovasc Diagn Ther* **3**, S246 (2017).
18. Zander, Z. K. & Becker, M. L. Antimicrobial and antifouling strategies for polymeric medical devices. *ACS Macro Lett* **7**, 16 (2018).
19. Maan, A. M. C., Hofman, A. H., de Vos, W. M. & Kamperman, M. Recent developments and practical feasibility of polymer-based antifouling coatings. *Adv Funct Mater* **30**, 200936 (2020).
20. Faustino, C. M. C., Lemos, S. M. C., Monge, N. & Ribeiro, I. A. C. A scope at antifouling strategies to prevent catheter-associated infections. *Adv Colloid Interface Sci* **284**, 102230 (2020).
21. Damodaran, V. B. & Murthy, N. S. Bio-inspired strategies for designing antifouling biomaterials. *Biomater. Res* **20**, 1 (2016).
22. Lowe, S., O'Brien-Simpson, N. M. & Connal, L. A., Antibiofouling polymer interfaces: poly(ethylene glycol) and other promising candidates. *Polym Chem* **6**, 198 (2015).
23. Goodman, S. L., Tweden K. S., & Albrecht R. M. Three-dimensional morphology and platelet adhesion on pyrolytic carbon heart valve materials. *Cells Mater* **5**, 15 (1995).
24. Damodaran, V. B. & Murthy, N. S. Bio-inspired strategies for designing antifouling biomaterials. *Biomater Res* **20**, 18 (2016).
25. Feldschuh, J. & Enson, Y. Prediction of the normal blood volume: Relation of blood volume to body habitus. *Circulation* **56**, 4 (1977).
26. De Buck, S. S. et al. Prediction of human pharmacokinetics using physiologically based modeling: A retrospective analysis of 26 clinically tested drugs. *Drug Metab Dispos* **35**, 10 (2007).
27. Biological Responses to Metal Implants – FDA, September (2019).
28. Davidovsky, A.G. The aging, biodegradation and toxic effects of the polymer biomedical implants. *Dental Practice* **869**, 259 (2020).
29. Eliaz, N. Corrosion of metallic biomaterials: A review. *Materials (Basel)* **12**, 407 (2019).
30. Manivasagam, G., Dhinasekaran, D., & Rajamanickam, A. Biomedical implants: Corrosion and its prevention - A review. *Recent Pat Corros Sci* **2**, 40, (2010).

Reviewers' Comments:

Reviewer #1:

Remarks to the Author:

Attached

The authors have provided an extensive and detailed response to the comments of the three reviewers. Each of the reviews recommended publication of the manuscript subject to major revisions.

Reviewer 1.

Details of th

Question 1.1: *One aspect that is unclear is how the current collectors are physically connected to the potentiostat? Details of this are not evident in the methods and need to be.*

This has been addressed and sufficient details are now included.

Question 1.2: *It's also unclear as to how the device is charged for use? The device clearly operates as a supercapacitor but e.g., in figure 4 c, are the capacitors charged prior to use?*

This has been addressed and sufficient details are now included

Question 1.3: *The properties of the devices in blood are enhanced in comparison with 0.9% saline, however the explanation for this enhancement is not clear. The text “Blood consists of enzymes and living cells that produce strong bioelectrocatalytic reactions)” line 107 is correct in itself but does not apply clearly here. The electrode layers in the devices are enclosed and only exposed at the edge (the side view of the swiss roll) as in fig 2h. Its not clear that bioelectrocatalytic reactions can occur as such reactions require an electrode surface that usually has to undergo modification to enable electrocatalytic reactions to occur (and by definition electrocatalytic implies electron transfer). Reactions such as those that utilise ADP/ATP do not involve electron transfer, typically this requires the use of NAD(P) cofactors and/or redox active enzymes. This aspect of the manuscript needs to be addressed and a more detailed explanation of the effect in blood needs to be provided before*

The response and additional experimental details are sufficient, however its not appropriate to describe the reactions as bioelectrocatalytic as the evidence for this is not presented (nor as strong bioelectrocatalytic reactions). It would be better to state “produce an electrochemical response” in place of “strong bioelectrocatalytic reactions”

Question 1.4: *An additional comment is that the manuscript would benefit from a discussion on the size of the devices in terms of the radius of the channel, ca 30 um.*

- a. *Was this radius selected or a consequence of the method of preparation?*
- b. *Can the size be altered and if so over what range?*
- c. *Does this size exclude cells (RBCs have a radius of ca 6 um and should be able to enter the channel).*

This has been addressed and sufficient details are now included.

Reviewer 2

Question 2.1: *As we know, the stability window in aqueous environment is only 1.21 V. The operation of the supercapacitor at 3 V is out of the stability window of the blood and may cause gas evolution. How is the device addressing this issue? Furthermore, as the BSC requires direct contact with the blood to maintain the voltage, how can three BSCs connect in series while sharing the same electrolyte?*

The response is adequate but text (single line) stating that SU8 was included to suppress water splitting should be included in the manuscript itself

Question 2.2: *The roll-up device occupies a significant amount of volume in the blood vessel, and can cause a significant increase in local blood pressure and may cause a reduction in flow rate. The authors propose to insert the device inside the blood vessels. Some comments on this concern would be helpful*

The response is sufficient but again, a short statement to the effect that significant vascular blockage (<30%) does not occur

Question 2.3: *How will the rolled-up BSC be charged while implanted within the blood vessel? Is it through a wired connection? If so, how to connect to the BSC through the blood vessel?*
This has been addressed.

Question 2.4: *The blood vessel dilates and contracts for tens to hundreds of times per minute, which might cause repeated mechanical deformations on the BSC device. Figure 3h have shown the stability of the BSC in 75 cycles of compression, but its stability through more deformation cycles are needed to ensure its mechanical resiliency.*

This has been addressed.

Question 2.5: *The abstract should contain no references. Please modify based on the format requirement.*

This has been addressed.

Question 2.6: *Is there any biofouling issue when applied to blood?*

This has been addressed in the response but needs to be included in the manuscript or SI

Question 2.7: *The authors claim that the supercapacitor performance was tested under blood flow in a PDMS microfluidic channel. How do they measure it? Do they fix the device or free the device in the flowing blood? Any mechanical force (e.g. crashing the channel walls) can influence the performance?*

This has been addressed and sufficient details are now included.

Question 2.8: *For the blood application, the authors should consider the biodegradability of the device and fate in the systemic circulation.*

This has been addressed.

Question 2.9: *What's the obstacles to moving forward to the in vivo trial? How to take out the device from the blood after use? Please specify the difficulties.*

This has been answered below, at this stage its not necessary to include the response in the manuscript.

Question 2.10: *Conclusions: can you elaborate more on the plan to translate this BSC system towards practical biomedical applications?*

This has been answered below, at this stage its not necessary to include the response in the manuscript.

Reviewer 3:

Question 3.1a: *1- The first conceptual limitation of this manuscript stems from the lack of device encapsulation: The authors reported that the blood components such as enzymes and cells boost the capacitive behavior by 40% due to bioelectrocatalysis at the surface of the current collector which is reasonable. However, the manuscript fails to realize the impact of oxidizing and reducing the components of the blood during thousands of charge-discharge cycles as part of the biosupercapacitor operations. During this process all redox based components of the blood such as redox enzymes, GSH/GSSG system, and many others will be significantly disrupted. The authors could visualize the damaging effects by applying the same 5000 charge-discharge cycles of their device to a "redox dye" and observe the color changes. I suggest the authors to run biochemical assays of the blood redox systems before and after 5000-100,000 charge-discharge cycles and discuss this effect in the manuscript to highlight the limitation of this device for a long-term use.*

This has been addressed in the response but details of the response should be included in the manuscript/SI..

Question 3.1b: *Lack of encapsulation will necessarily lead to biofouling problems when the device is implanted for a period of days to weeks. The adsorption nature of many the whole blood components on metal surfaces (current collector) which is further maximized by the depositions of blood components due to charge-discharge cycles will eventually block the hollow tube of the “swiss-roll” device and compromise it’s performance. Biofouling and it’s impact should be assessed and discussed in the manuscript.*

See response to 2.6 above.

Question 3.1c: *The authors charged the biosupercapacitor up to 1.6 V while blood is in contact with the current collector. This voltage will lead to a degree of electrochemical water splitting (90% of human blood is water). Please review OER and HER reactions of gold (current collector) in neutral media. It is important not charge the biosupercapacitor to a higher potential window than water splitting reactions to avoid damage. Please consider reducing the voltage, or justifying the safety of applying 1.6 V to blood.*

See response to Reviewer 2 above

Question 3.2: *Unlike batteries and fuel cells, the reported biosupercapacitor can not work alone and requires momentary charging source for continuous charging so that the biosupercapacitor can be practical. The authors in this work charged the biosupercapacitor from an outside source to use it as a power source for pH sensing. The authors need to discuss in their manuscript a real-life scenario of the source of power to charge such a biosupercapacitor to realize a functional power source for implantable device. Would this intended to be hybrid with a nanogenerator, a biofuel cell, etc. This discussion should be added to introduction and other parts of the manuscript as relevant. This will give the reader a direction of how third biosupercapacitor will be used in real implantation scenario.*

This has been addressed and sufficient details are now included.

Question 3.3: *The authors monitored the pH of the blood as a function of the capacitance change with a reasonable sensitivity. However, the selectivity of the sensor may not hold. In other words, the pH sensing was tested in DI water-based electrolyte at different pH, however, many factors can affect the*

capacitance of the biosupercapacitor if the test is to be performed in a complex media such as the blood. The authors should provide the sensitivity of this sensor in blood or at least plasma and discuss the results in the manuscript.

This has been addressed and sufficient details are now included.

Question 3.4: *The word supercapacitor in essence is a commercial terminology and not a scientific one even though it is commonly used in publications. This “super” terminology was used to indicate that the device stores more energy. The correct scientific name of the device is electrochemical capacitor, in this case, bioelectrochemical capacitor. I suggest the author to mention the word biosupercapacitor may be once and refer to the device as bioelectrochemical capacitor elsewhere in the MS.*

This has been addressed.

Question 3.5: *I suggest the author to add the exact length and diameter of their biosupercapacitor to the text.*

This has been addressed and sufficient details are now included.

Comment 3.7: *The authors should cite and discuss relevant biosupercapacitor literature such as Mosa, I. M.; Pattammattel, A.; Kadimisetty, K.; Pande, P.; El-Kady, M. F.; Bishop, G. W.; Novak, M.; Kaner, R. B.; Basu, A. K.; Kumar, C. V.; Rusling, J. F., Ultrathin Graphene–Protein Supercapacitors for Miniaturized Bioelectronics. *Advanced Energy Materials* 2017, 7 (17), 1700358. Pankratov, D.; Shen, F.; Ortiz, R.; Toscano, M. D.; Thormann, E.; Zhang, J.; Gorton, L.; Chi, Q., Fuel-independent and membrane-less self-charging biosupercapacitor. *Chemical Communications* 2018, 54 (83), 11801-11804.*

Response 3.7: References have been added at appropriate locations.

Reviewer #2:

Remarks to the Author:

The authors have nicely revised their manuscript in response to most of my questions and comments. In regards to comment 2.3 about the practical biomedical applications, I hope the authors can address this in their future work. Overall, these changes greatly improved the quality towards publication in Nature Communications.

Reviewer #3:

Remarks to the Author:

The authors did an excellent job addressing the reviewer comments. The authors made a necessary change to their device structure by passivating the exposed current collector layer and isolate it from the surrounding blood or biofluids and thus avoiding most of the drawbacks that were highlighted earlier in the reviewer comments regarding splitting the water content of the blood, etc. In addition, the authors added new experiments to showcase that their pH sensor could work in complex media such as serum. Although the sensitivity of the sensor is quietly impacted, this drop in sensitivity is expected when the testing is shifted to a more complex and representative sample such as serum or whole blood. This work is very interesting and I recommend our esteemed editors to accept the current revised version of this manuscript as is.

Sincerely

Islam Mosa

Concerning Question #1.3:

Comment reviewer #1: The response and additional experimental details are sufficient, however its not appropriate to describe the reactions as bioelectrocatalytic as the evidence for this is not presented (nor as strong bioelectrocatalytic reactions). It would be better to state “produce an electrochemical response” in place of “strong bioelectrocatalytic reactions”.

We agree with the reviewer and rephrased the text at all relevant positions (marked in blue in the revised manuscript).

Concerning Question #2.1:

Comment reviewer #1: The response is adequate but text (single line) stating that SUB was included to suppress water splitting should be included in the manuscript itself.

Although this was a question originally raised by reviewer #2, we think it's a minor point and decided to add the sentence “The SU-8 was introduced to suppress water splitting.” into the manuscript (marked in blue on p. 8).

Concerning Question #2.2:

Comment reviewer #1: The response is sufficient but again, a short statement to the effect that significant vascular blockage (<30%) does not occur.

The manuscript includes the following sentences: “The target vessel must be selected such that the nBSC does not block the vessels by more than 30%. Any blockage over 30% would cause a dramatic decrease in blood flow velocity and increase in local blood pressure leading to vascular remodelling and stenosis”

Concerning Question #2.6:

Comment reviewer #1: This (the biofouling) has been addressed in the response but needs to be included in the manuscript or SI.

Although this was a question originally raised by reviewer #2, we think it's a minor point and decided to add the sentence “Although biofouling was not observed around the nBSCs during the short term (5.5 days) in-flow GCD cycling, anti-biofouling strategies over the whole operation lifetime need to be implemented in future application scenarios to prevent vascular blockage.” into the manuscript (marked in blue on p. 5).

Concerning Question #2.9:

Comment reviewer #1: This (the difficulties in moving ahead with in-vivo trials) has been answered below, at this stage its not necessary to include the response in the manuscript.

This was a question originally raised by reviewer #2. We responded to the question and revised our manuscript accordingly in our previous resubmission. Reviewer #2 accepted our revision. We therefore keep the text as is.

Concerning Question #2.10:

Comment reviewer #1: This (the plan for translation towards biomedical applications) has been answered below, at this stage its not necessary to include the response in the manuscript.

This was a question originally raised by reviewer #2. We responded to the question and revised our manuscript accordingly in our previous resubmission. Reviewer #2 accepted our revision. We therefore keep the text as is.

Concerning Question #3.1a:

Comment reviewer #1: This has been addressed in the response but details of the response should be included in the manuscript/SI.

This was a question originally raised by reviewer #3. We responded to the question, provided all details in the manuscript/SI and revised our manuscript accordingly in our previous resubmission. Reviewer #3 accepted our revision. We therefore keep the text as is.